ecology, health and disease and epidemiology

*Anopheles* mosquitoes, malaria ecology, population dynamics, seasonality, epidemiology

**Author for correspondence:**
Charles Whittaker
e-mail: charles.whittaker16@imperial.ac.uk

# A novel statistical framework for exploring the population dynamics and seasonality of mosquito populations

Charles Whittaker[1], Peter Winskill[1], Marianne Sinka[2], Samuel Pironon[3], Claire Massey[4], Daniel J. Weiss[5,6], Michele Nguyen[7], Peter W. Gething[5,6], Ashwani Kumar[8], Azra Ghani[1] and Samir Bhatt[1,9]

[1]MRC Centre for Global Infectious Disease Analysis, School of Public Health, Imperial College, London, UK
[2]Department of Zoology, University of Oxford, Oxford, UK
[3]Royal Botanic Gardens Kew, Richmond, London, UK
[4]Big Data Institute, University of Oxford, Old Road Campus, Oxford, UK
[5]Malaria Atlas Project, Telethon Kids Institute, Perth Children's Hospital, Nedlands, WA 6009, Australia
[6]School of Public Health, Curtin University, Bentley, WA 6102, Australia
[7]Asian School of the Environment, Nanyang Technological University, Singapore, Singapore
[8]Vector Control Research Centre, Indira Nagar, Puducherry, India
[9]Section of Epidemiology, Department of Public Health, University of Copenhagen, Copenhagen, Denmark

CW, 0000-0002-5003-2575; MN, 0000-0001-9017-1978

Understanding the temporal dynamics of mosquito populations underlying vector-borne disease transmission is key to optimizing control strategies. Many questions remain surrounding the drivers of these dynamics and how they vary between species—questions rarely answerable from individual entomological studies (that typically focus on a single location or species). We develop a novel statistical framework enabling identification and classification of time series with similar temporal properties, and use this framework to systematically explore variation in population dynamics and seasonality in anopheline mosquito time series catch data spanning seven species, 40 years and 117 locations across mainland India. Our analyses reveal pronounced variation in dynamics across locations and between species in the extent of seasonality and timing of seasonal peaks. However, we show that these diverse dynamics can be clustered into four 'dynamical archetypes', each characterized by distinct temporal properties and associated with a largely unique set of environmental factors. Our results highlight that a range of environmental factors including rainfall, temperature, proximity to static water bodies and patterns of land use (particularly urbanicity) shape the dynamics and seasonality of mosquito populations, and provide a generically applicable framework to better identify and understand patterns of seasonal variation in vectors relevant to public health.

## 1. Background

With over 400 000 estimated deaths in 2019 [1], malaria represents one of the most serious infectious diseases globally [2]. Nineteen countries in sub-Saharan Africa along with India account for almost 85% of the global burden [3], with *Plasmodium falciparum* most prevalent in African settings, and India alone accounting for almost 50% of the global *Plasmodium vivax* burden [4]. Transmission occurs via mosquito vectors belonging to the *Anopheles* genus—these vectors are heterogeneously distributed across the globe [5,6], with marked differences in the transmission dynamics of malaria across different ecological contexts.

Much work has focussed on characterizing the global spatial distribution (presence/absence) of these malaria vectors (and other mosquitos relevant to public health) [7,8]. This work represents a vital input to surveillance and control programmes aimed at mitigating the impacts of vector-borne diseases

worldwide. By contrast, beyond a focus on dynamics in single locations, comparatively less attention has been paid to understanding the temporal patterns of vector abundance, and how these dynamics are shaped by the local environment. Mosquito populations are highly temporally dynamic, exhibiting substantial annual fluctuations in size [9,10] that drive the temporal profile of disease risk. Understanding the determinants of these dynamics is important given that the efficacy of many interventions including seasonal malaria chemoprevention [11,12] and indoor residual spraying [13,14] depends on the timing of their delivery in relation to seasonal peaks in risk. Effective utilization of these interventions will be vital for achieving the goals of the World Health Organization's 'high burden, high impact' strategy, which aims to substantially reduce/eliminate malaria in India and the ten African nations with the highest global burden [15].

Many questions remain surrounding the drivers of mosquito population dynamics. A close relationship has been observed between rainfall occurrence, peaks in mosquito populations and malaria cases [16], including for *Anopheles gambiae s.l.* [17–19] across African settings and *Anopheles dirus s.l.* across India and southeast Asia [20], in keeping with the preferences some species display for transient, rain-fed pools of water as larval habitats [21]. However, a number of studies including work by Cohuet *et al.* [22] and Mendis *et al.* [23] on *Anopheles funestus s.l.* populations have identified varying degrees of seasonality, including population abundance peaking in the dry season [24]. Relatedly, a number of studies have shown *Anopheles annularis s.l.* present in significant numbers over the course of the entire year, despite highly seasonal rainfall [10,25,26]. This brings into question how generalizable relationships between rainfall and mosquito population dynamics are. The influence of other factors such as temperature (which influences many mosquito traits including larval development [27], biting rates and mortality rates [28]) remains similarly unclear. Recent work has suggested that considerations of both rainfall and temperature are necessary to understand seasonal patterns of malaria incidence [29]. However, these analyses have been restricted to a small number of settings across sub-Saharan Africa; leaving the influence of temperature on mosquito population dynamics largely unexplored in other settings. Previous work has also suggested a role for a number of other ecological factors beyond temperature and rainfall in shaping mosquito population dynamics. These include land use, such as irrigative practices [30] or structure of the built-environment in urban settings [31], as well as the local hydrological environment and presence of long-lived water bodies [32,33], which potentially provide viable habitats for mosquito larvae year-round.

While numerous studies of anopheline seasonality have been carried out, the focus is typically on a single species and/or location—such studies are rarely synthesized together to identify generalizable patterns and facilitate systematic comparisons across key vector species. Here, we develop a novel statistical framework enabling characterization of the temporal patterns displayed by mosquito species complexes and identification of 'dynamical archetypes' sharing similar temporal properties. Using mainland India as a case study, we collate temporally disaggregated mosquito catch data from across the country and use this statistical framework to better understand variation in mosquito population dynamics and the factors underlying this variation. Our work reveals pronounced heterogeneity in the extent and nature of seasonal dynamics, both between species complexes and across different locations. In doing so, our results highlight the importance of considering both species composition and ecological structure when implementing interventions aimed at controlling vector-borne diseases.

## 2. Methods

### (a) Systematic review of Indian entomological literature

Web of Science and PubMed databases were searched on 17 October 2017 using the keywords 'India' AND 'Anophel*' to identify references with temporally disaggregated entomological data. We identified 1945 records with 1556 remaining after removing duplicates. Following Title and Abstract screening, 281 records were retained for full text evaluation. We included records containing temporally disaggregated adult mosquito catch data with monthly (or finer) temporal resolution spanning at least 12 months that had not been conducted as part of vector control intervention trials, and where sufficient information to geolocate the catch site was provided. We retained 78 references that yielded 117 geolocatable areas across India. These references contained 272 time series spanning the malaria vectors *Anopheles annularis s.l., culicifacies s.l., dirus s.l., fluviatlis s.l., minimus s.l., stephensi s.l.* and *subpictus s.l.* and spanning five collection methods. See electronic supplementary material information for further details.

### (b) Time series fitting and interpolation

To smooth the noise in the mosquito catch data, we fitted a Gaussian Process model to each of the extracted time series, using a Negative Binomial likelihood to account for overdispersion in the data

$$\theta, \sigma \sim \pi(\theta, \sigma)$$
$$\text{and} \quad f \sim \text{GP}(0, K_\theta(x))$$
$$y_i \sim \text{negative binomial } (e^{f(x_i)}, \sigma) \, \forall \, i \in \{1, \dots, N\},$$

where $f$ is a distribution of functions from a zero-mean Gaussian process with covariance $K_\theta$, $f(x)$ are function evaluations at times $x$, $y$ are the mosquito counts indexed by time $i$, and $\sigma$ and $\theta$ are hyperparameters defining the overdispersion of the negative binomial distribution and functional form of the covariance function. Mosquito dynamics are typically characterized by repeating patterns occurring either seasonally or annually, a periodic kernel function was therefore used to define the covariance between pairs of points

$$k(x, x') = \alpha^2 \exp\left(\frac{-2}{l^2} \sin^2\left(\frac{\pi|x - x'|}{p}\right)\right),$$

where $p$ is the period over which we would expect points to show similar dynamics (i.e. period of 12 would imply points separated by 12 months to be most similar), $\alpha$ specifies the magnitude of the covariance, and $l$ a length-scale parameter constraining the extent to which two values can covary. Weakly informative priors were used although the results were not sensitive to the choice of prior (see electronic supplementary material, figure S4). Fitting was undertaken using the programming language STAN [34].

### (c) Time series characterization and clustering by features

Motivated by previous work providing a framework to statistically characterize the empirical structure of time series data [35] and seasonality of malaria case incidence [36], we calculated summary statistics for each smoothed time series to characterize their temporal properties. These include the Kullback–Liebler divergence

(measuring the divergence of the time series from a uniform distribution), the median of the period ($p$) from the negative binomial Gaussian process fitting (informing the dominant temporal modality present), the proportion of points greater than 1.65× the mean (measuring how peaked the time series is), the distance of the first peak from January, and then three features arising from fitting one- and two-component Von Mises distributions to the smoothed time series: the mean of the one-component Von Mises, the number of peaks (determined by comparing quality of fit for one- and two-component Von Mises distributions), and weight ($\omega$), specifying the comparative contributions of each component in the two-component fitting. See electronic supplementary material information for further details. We then applied a principal components analysis to these results to identify a lower-dimensional representation of the structure present in the data and implemented $k$-means clustering to identify clusters of time series with similar temporal features (i.e. this clustering assigns each smoothed time series to one cluster).

### (d) Statistical modelling and prediction of seasonal modality

For each study location, we extracted a suite of environmental variables derived from satellite data. These include time-period and location specific rainfall data from The Climate Hazards Group Infrared Precipitation With Stations (CHIRPS) dataset [37], BioClimatic variables (defined from monthly rainfall and temperature satellite data [38]), various measures of aridity [39,40], covariates describing the seasonality and extent of water bodies [41], land cover [42] and a number of other variables previously used to define the distribution of anopheline vectors [43]. A complete list of these is in electronic supplementary material, table S2. These covariates (25 in total) and a covariate for anopheline species (one for each time series indicating which species it belonged to) were used in a regularized multinomial logistic regression model predicting the cluster (of time series with similar temporal properties, assigned based on the results of the $k$-means clustering) a particular time series belonged to. Fitting this model yielded regression coefficients describing the strength of association between a species complex/environmental variable and membership of a particular cluster—specifically, one coefficient per cluster and predictor. We then integrated these results with recently produced maps of vector presence/absence as part of work conducted with the Humbug Project (http://humbug.ac.uk; funded through a Google Impact Challenge grant) to generate predictive maps of mosquito dynamics across India (see electronic supplementary material information).

## 3. Results

### (a) Substantial diversity in mosquito population dynamics within and between species

We identified 272 time series from 117 locations across India through the systematic review, spanning seven species complexes that represent the country's dominant malaria vectors (electronic supplementary material, figure 1A). These noisy time series (electronic supplementary material, figure S1) were then smoothed using a negative binomial Gaussian process-based framework (figure 1b). Substantial variation in temporal dynamics was observed between different species complexes in the degree and timing of seasonality. While *Anopheles dirus s.l.* populations tended to peak during the monsoon period (typically June to September), many *Anopheles fluviatilis s.l.* populations peaked between November and February (the dry season across most of India), reaching their lowest density during the monsoon. *Anopheles*

*dirus s.l.* populations demonstrated the highest degree of seasonality with an average of 75% of the total annual catch being concentrated in a four month time-period (figure 1c). For *Anopheles annularis s.l.*, only 53% of the total annual catch on average was caught in any four month period. In addition to this variation between species complexes, we also observed extensive variation in temporal dynamics within a species complex. Across the 85 time series collated for *Anopheles culicifacies s.l.*, populations varied substantially in both the extent and timing of their seasonal peaks; this ranged from sharp peaks in the monsoon season to less seasonal characteristics similar to those observed for *Anopheles annularis s.l.* A range of dynamics was also observed for time series belonging to *Anopheles stephensi s.l.*, from peaks coincident with the monsoon season to bimodal dynamics displaying peaks both during and outside the rainy season.

### (b) Distinct temporal 'archetypes' in mosquito population dynamics

An array of summary statistics were calculated for each time series to characterize their temporal properties (see electronic supplementary material, methods and figure S2). This was followed by $k$-means clustering of the results, to delineate the observed variation into discrete groups, each characterized by distinct temporal patterns. We identified four groups (figure 2a)—these included time series peaking during the monsoon season (Cluster 1), displaying bimodal characteristics (Cluster 2), peaking in the dry season (Cluster 3) or displaying more perennial patterns of abundance (Cluster 4) (figure 2c). Cluster assignment was robust to choice of prior used in the time series smoothing (electronic supplementary material, figure S3). Average catch size varied between clusters, ranging from a median catch size of 356 for Cluster 2 to 42 for Cluster 4 (see electronic supplementary material, figure S4). The distinct patterns displayed by each group were not due to differences in the timing and extent of rainfall across India—we collated location and time-period specific rainfall data for each study (from the CHIRPS dataset [37]) and calculated the cross-correlation between catch and rainfall. This varied between clusters—a high positive cross-correlation was observed for Cluster 1 (average $r = 0.52$), but a negative correlation for Cluster 3 ($r = -0.41$) and low correlation for Clusters 2 and 4 ($r = -0.08$ and 0.03, respectively, figure 2b; electronic supplementary material, figure S5). This suggests that the observed patterns represent genuine differences in how mosquito populations respond to rainfall. For some species complexes, the majority of their time series belonged to a single cluster (figure 2d)—*Anopheles dirus s.l* time series were restricted primarily to Cluster 1 (monsoon season peaking) while *Anopheles fluviatilis s.l.* time series were almost exclusively found in Cluster 3 (dry season peaking). By contrast, time series belonging to *Anopheles culicifacies s.l.* appeared across all four clusters—indicating either that different sibling species within the complex display distinct temporal dynamics or that mosquito populations belonging to the species complex are able to adopt a diverse array of temporal dynamics depending on the particular ecological setting.

### (c) Mosquito population dynamics are driven by a complex interplay of abiotic and biotic factors

Using binary indicators for species complex (seven total, indicating which species complex a particular time series belongs

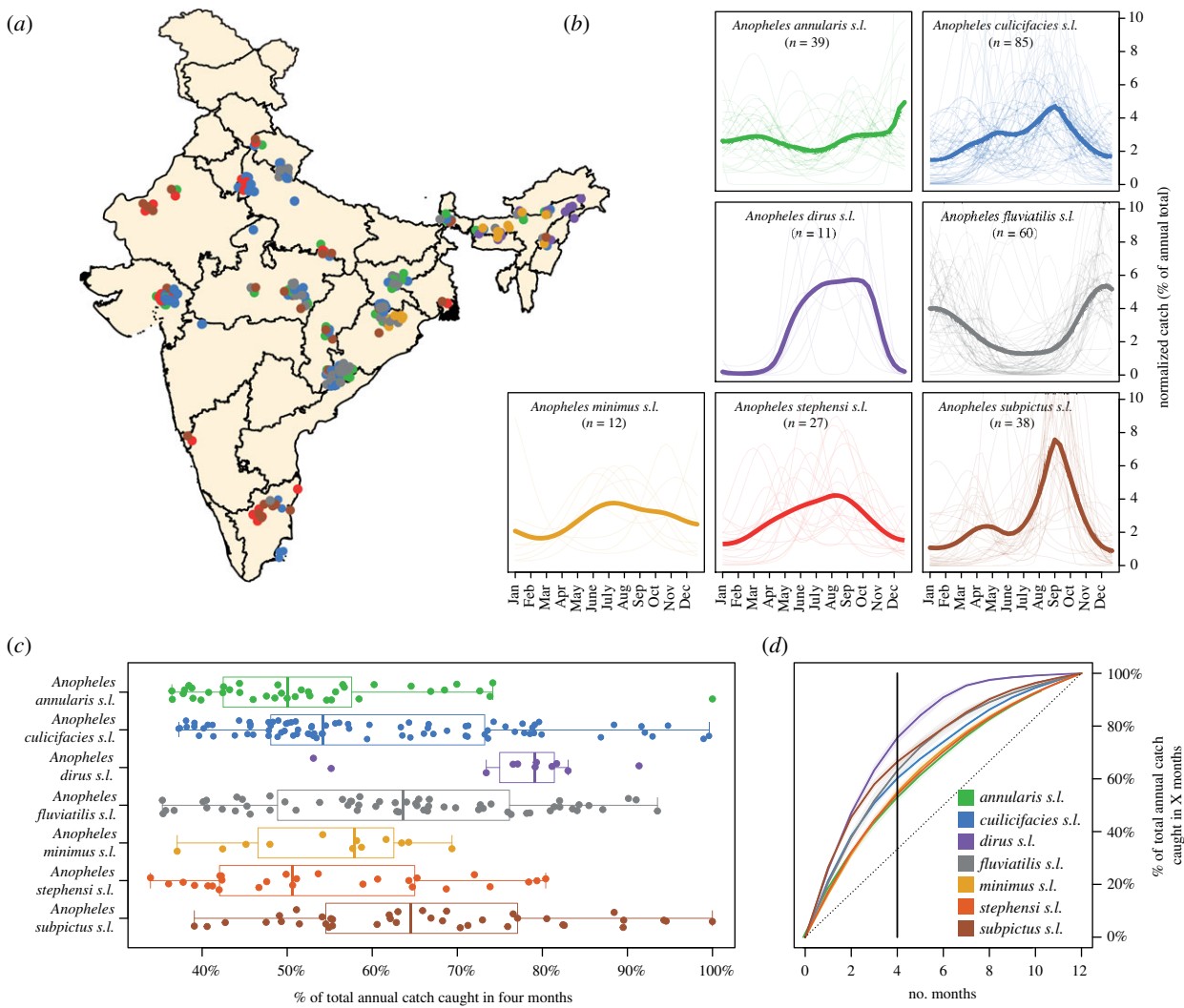

**Figure 1.** Exploring species complex-specific patterns of mosquito population dynamics. (*a*) Map of India showing the different locations for which time series data was available. Points represent a single collected time series, coloured according to species. (*b*) Normalized, Gaussian process fitted time series disaggregated by species complex. Pale lines represent a single time series for that particular species complex, and the brighter line is the mean of all of the time series belonging to that species complex, evaluated at that particular timepoint. (*c*) Boxplot of the maximum percentage of total annual study catch caught in any consecutive four month period (with a higher value implying greater seasonality). Each point is a study, coloured according to anopheline species. (*d*) Mean percentage of the total annual study catch caught across different months for each species. (Online version in colour.)

to) and a suite of ecological variables, we fitted a multinomial logistic regression model to the cluster labels (i.e. which cluster each time series had been assigned to) to explore potential factors underlying the observed variation in temporal dynamics. This framework produces one coefficient estimate for each cluster and predictor (a total of four coefficients per cluster and predictor), with that coefficient defining the strength of the association between a predictor and a particular cluster.

Across the species complex coefficients, *Anopheles culicifacies sl.* and *Anopheles subpictus s.l.* demonstrated positive associations with Cluster 1 (monsoon peaking dynamics), whereas this relationship was negative for *Anopheles fluviatilis s.l.* (the species-complex associated with Cluster 3 instead) and *Anopheles annularis s.l.* was most strongly associated with Cluster 4 (perennial dynamics). We employed a hierarchical clustering approach to identify groups of species with similar patterns of association with specific temporal dynamics (figure 3*a*). *Anopheles culicifacies s.l.* and *Anopheles subpictus s.l.* clustered together and showed a positive association with Cluster 1 and a negative association with Cluster 3. *Anopheles fluviatilis s.l.* clustered on its own,

positively associated with Cluster 3 and negatively associated with Cluster 1. There were disparities in the number of time series for each species (85 for *Anopheles culicificaes s.l.* versus 11 for *Anopheles dirus s.l.*) and so we explored how robust the results of this clustering were to subsampling the data so that all species had the same number of time series (as *Anopheles dirus s.l.*). These groupings were robust to subsampling, except in the case of *Anopheles dirus s.l.*, which clustered with *Anopheles culicifacies s.l.* and *Anopheles subpictus s.l.* (showing positive associations with Cluster 1 dynamics, and a negative association with Cluster 3 dynamics, electronic supplementary material, figure S6).

Temperature seasonality and total annual rainfall were strongly associated with Cluster 1 (which possessed the dynamics most strongly correlated with rainfall, electronic supplementary material, figure S5) (figure 3*b*). By contrast, perennial dynamics (Cluster 4) strongly associated with the presence of water bodies and negatively associated with temperature seasonality and rain seasonality. Strong associations with land cover were observed for Cluster 2 (strongly negative for urbanicity) and Cluster 3 (strongly positive for woody savannas). We ranked the coefficients

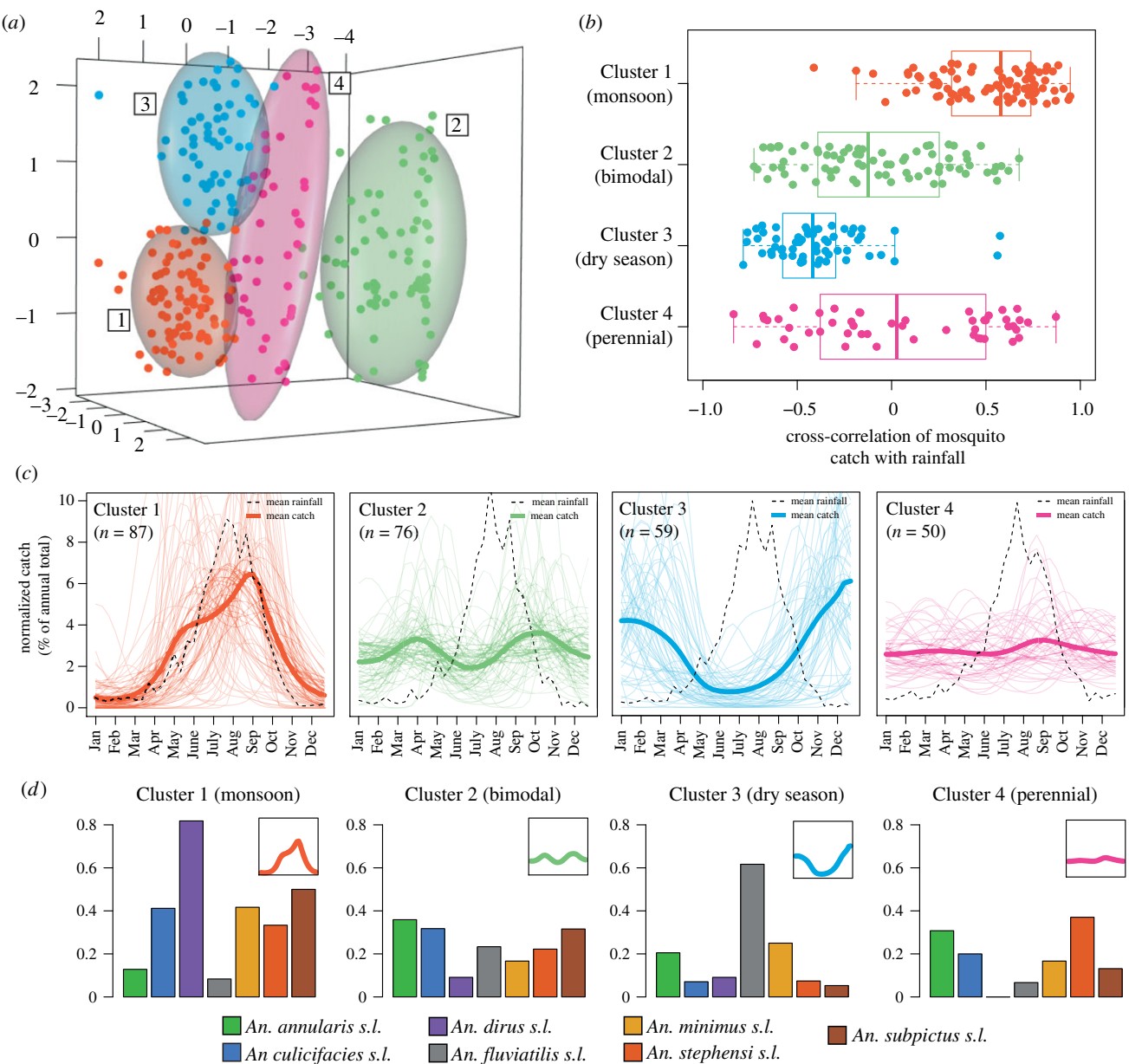

**Figure 2.** Characterization and clustering of time series with similar temporal properties. (*a*) Results of PCA and *k*-means clustering for four clusters. Point colour refers to cluster membership, ellipsoids demarcate the 75th quantile of the density associated with each cluster. (*b*) Boxplot of cross-correlation between rainfall and mosquito catch for each location and time series. Rainfall data from the CHIRPS dataset [37] is specific to study location and time period. Each point indicates an individual time series. (*c*) Time series belonging to each cluster. Pale lines represent individual time series, brighter line the mean of all the time series belonging to that cluster. Dashed black line is the mean rainfall for locations belonging to the cluster. (*d*) Proportion of time series for each species complex belonging to each cluster; bars indicate different species complexes and *y*-axis the proportion of time series (for a given species complex) belonging to that cluster. (Online version in colour.)

for each environmental variable within each cluster according to their magnitude, and selected the 15 with the strongest association in each cluster (positive or negative). The top 15 variables for each cluster were then compared to assess the extent of overlap, revealing that each cluster tended to associate with a unique set of ecological factors (figure 3*c*), and an analysis of the correlation of all coefficient values between clusters revealed them to be highly negatively correlated (electronic supplementary material, figure S7).

### (d) Predictive mapping highlights the extensive variation in mosquito dynamics across India

We integrated these results with spatial predictions of mosquito species complex presence/absence to produce predictive maps of mosquito population dynamics; specifically, to generate estimates of the probability that a given location contains greater than or equal to 1 mosquito species complex displaying a particular temporal pattern. These results (a probability per pixel) were thresholded arbitrarily at 0.67 to produce a binary indicator (i.e. a value of 1 if the probability for a given pixel is greater than 0.66 and 0 otherwise) and the results presented in figure 4 (see electronic supplementary material, figure S9 for raw, non-arbitrarily thresholded probabilities). Our results predict that monsoon peaking dynamics (Cluster 1) are most likely in the north and northeast (figure 4*a*). This contrasts with the predicted spatial distribution of bimodal dynamics (Cluster 2), which are more likely across central India. Dynamics involving peaks during the dry season tracks the predicted spatial

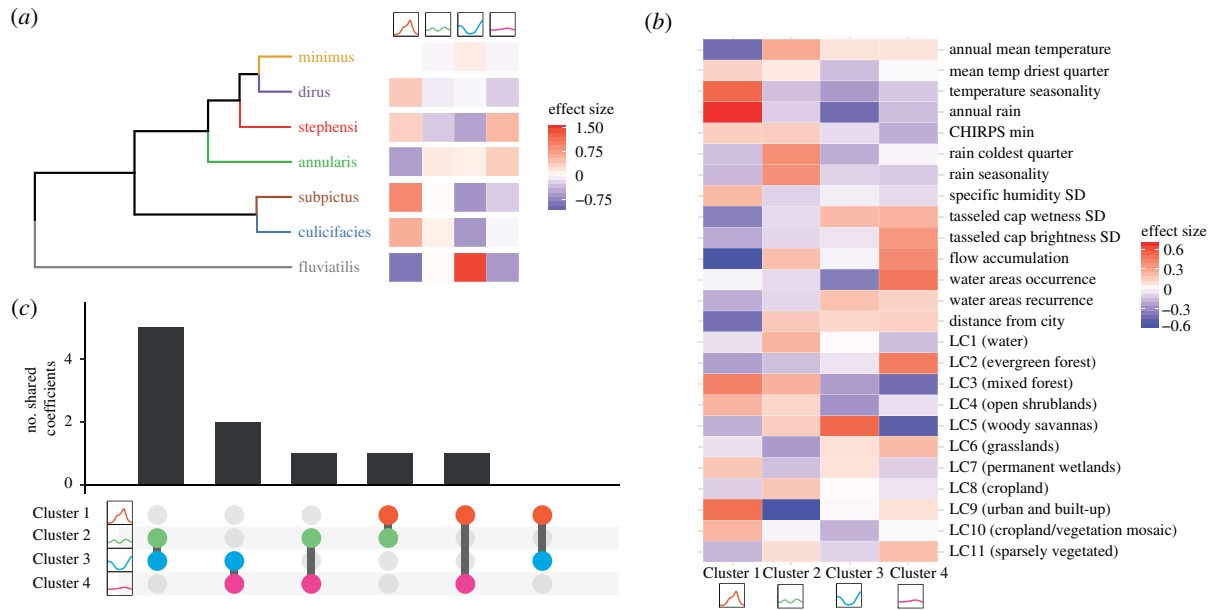

**Figure 3.** Exploring drivers of mosquito population dynamics using multinomial logistic regression. (*a*) Hierarchical clustering of the regression results for each species complex, as defined by the set of coefficient values describing the strength of the association between that species complex and the particular cluster. (*b*) The strength of the association between each of the 25 environmental covariates used and the relevant temporal cluster. (*c*) Upset plot summarizing the top 15 environmental variable coefficients associated with each cluster. The *x*-axis indicates the specific pairwise cluster comparison, *y*-axis the number of shared top 15 covariates between the two clusters. (Online version in colour.)

distribution of *Anopheles fluviatilis s.l.* closely and are pre-dicted to be most probable across central India (figure 4*c*)—a similar pattern was observed for spatial predictions of per-ennial dynamics (figure 4*d*). Together these results suggest that spatial variability in both species complex occurrence and environmental factors together generate the complex pat-terns of mosquito temporal dynamics observed across India.

## 4. Discussion

Understanding the temporal dynamics of malaria trans-mission represents an important input to effective deployment of control interventions. Here, we develop a stat-istical framework enabling systematic evaluation and comparison of temporal dynamics and patterns of seasonality in mosquito populations, and apply this framework to a col-lection of temporally disaggregated mosquito time series catch data from across India to explore the dynamics. Our results reveal extensive variation in mosquito population dynamics between species complexes and across locations, ranging from highly seasonal and rainfall-concordant dynamics through to perennial and rainfall-discordant dynamics. Analysis of this variation has revealed a complex interplay between species complex-specific drivers and the broader ecological structure of the environment in shaping these dynamics.

In a manner largely independent of the ecological setting, *Anopheles fluviatilis s.l.* populations typically peaked during the dry season. While previous work has identified these dynamics [44,45], our work highlights the consistency of this observation across locations, showing that these dynamics are largely restricted to *Anopheles fluviatilis s.l.* and highlight the capacity for the population dynamics of a regionally important malaria vector to significantly depart from local patterns of rainfall. These results align with pre-vious work that has indicated streams and surrounding

stagnant water as larval habitats for this species complex [46]—such sites are typically unsuitable during the monsoon season when flooding occurs but become increasingly suit-able as the dry season ensues. By contrast, *Anopheles culicifacies s.l.* displayed a wide array of temporal dynamics depending on the sampling site, with different dynamics observed in different locations. These ranged from peaking during the monsoon to bimodal and even perennial behav-iour—a finding consistent with documented variation in the aquatic larval habitats used by the species complex [47–49]. An important limitation to note however is our inability to disaggregate time series according to sibling species, which frequently show differences in preferred types of aquatic larval habitat [48]. It, therefore, remains unclear whether the variation in temporal dynamics for *Anopheles culicifacies s.l.* is due to sibling species displaying distinct temporal dynamics or because *Anopheles culicifacies s.l.* temporal dynamics are generally more plastic than *Anopheles fluviatilis s.l.* (where the same dynamics were typically observed irre-spective of the broader ecological structure).

Our results highlight the limited utility of considering rainfall alone when trying to understand temporal patterns of mosquito abundance, with variable associations with rain-fall observed across the populations studied here. Indeed, we identified a significant impact of temperature on population dynamics, with temperature seasonality strongly positively associated with the highly seasonal, monsoon peaking seaso-nal dynamics (Cluster 1) and both temperature seasonality and rainfall seasonality negatively associated with perennial (Cluster 4) dynamics. The role of temperature in shaping mosquito population dynamics is increasingly being recog-nized [29,50], due in part to the significant influence it has on many individual mosquito life-history traits [27,28,51], including biting rate, fecundity and mortality (among others). The influence of temperature on these traits is typically nonlinear and unimodal with clear optima [52] and subject to interactions with other factors such as the demographic

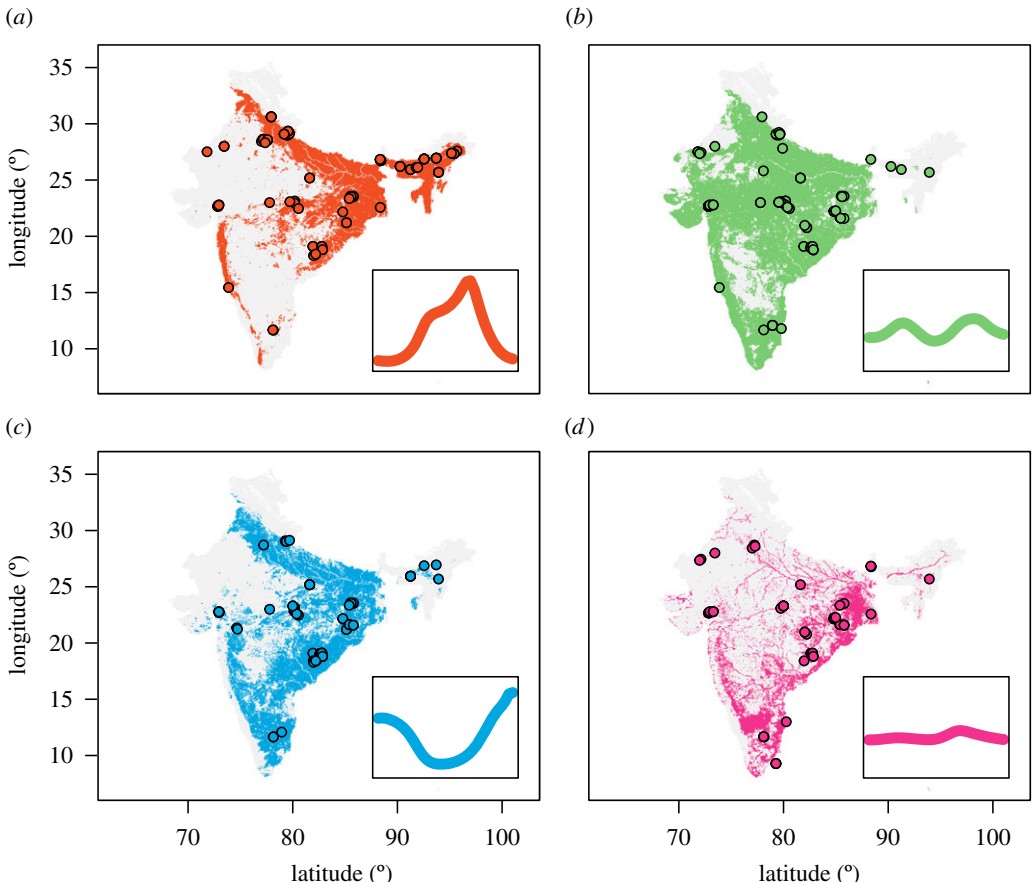

**Figure 4.** Predictive maps of mosquito population seasonality across India. The results of the multinomial logistic regression were integrated with maps describing the probability of presence/absence for different species complexes (not shown). These were used to generate estimates of a given area possessing at least one mosquito species complex with a particular temporal profile (as defined by the previously described clusters), with these probabilities then thresholded arbitrarily at 0.67 to produce a binary indicator (i.e. value 1 if the probability for a given pixel is greater than 0.66 and 0 otherwise—see electronic supplementary material, figure S9 for the raw, non-arbitrarily thresholded probabilities). Results of this are shown for (*a*) Cluster 1, (*b*) Cluster 2, (*c*) Cluster 3 and (*d*) Cluster 4. In all cases, coloured points indicate locations where a mosquito species complex displaying temporal dynamics belonging to that cluster were empirically observed. (Online version in colour.)

structure of the mosquito population [53]. This has significant consequences for mosquito population dynamics and, in turn, the range and dynamics of vector-borne diseases (such as malaria) they underpin [54–56]. Our results therefore suggest a role for both rainfall and temperature in shaping annual patterns of mosquito abundance and underscores the importance of considering seasonal fluctuations in a range of environmental variables when trying to understand seasonality in mosquito population dynamics.

In addition to temperature, we observed associations between temporal dynamics and a variety of other ecological covariates. The perennial patterns of abundance observed for Cluster 4 were strongly associated with flow accumulation and water area occurrence (acting as proxies for proximity to rivers and bodies of water). These factors were negatively associated with all other clusters. This is consistent with reports indicating that static water sources may provide viable sites for mosquito larvae year round [32,57] and highlights the importance of the local hydrological environment (which in the cases of large bodies of water is only partially dependent on patterns of rainfall) in shaping mosquito population dynamics. We also observed a significant influence of land cover patterns: urbanicity (measured by the two covariates land cover and distance to city) was consistently and positively associated with monsoon peaking dynamics (Cluster 1) and negatively associated with other clusters. This is

possibly due to the diverse array of physical features present in cities (e.g. tyres, wells, overhead tanks etc.) that are able to hold water following rainfall, and which have previously been characterized as larval habitats for a range of mosquito species [31,58]. Together, these results highlight that a broad array of ecological factors, including temperature, rainfall, the local hydrological environment and land-use structure among many others, influence the temporal dynamics of mosquito populations.

It is important to note that factors other than mosquito dynamics are also involved in defining the temporal profile of malaria risk. While an association between the size of mosquito populations and case numbers is well established [59,60], the nature of this relationship remains less clear. Interactions between malaria endemicity [61], mosquito abundance [62] and vector competence [29] can lead to nonlinear dynamics that can be further modified by human behavioural factors such as migration or occupational practices [63]. Due to limitations on the extent of entomological data describing relevant malaria metrics such as sporozoite positivity, we were unable to explore many of these factors. Similarly, the lack of disaggregation according to sibling species (which vary markedly in malaria vectorial efficiency) and accompanying epidemiological information (on malaria prevalence or incidence) precludes us from better resolving the comparative contributions of different mosquito species

to transmission. This limits our ability to translate temporal patterns of mosquito populations into relevant metrics such as the entomological inoculation rate (EIR). While we mitigate this limitation somewhat by focusing our analyses specifically on dominant vector species-complexes previously established as relevant to malaria transmission in India [64], it is not necessarily the case that each mosquito species analysed here is equally relevant to malaria transmission. Future work integrating these analyses with those exploring seasonality of case incidence (cf. [36]) would therefore probably prove instructive.

There are a number of limitations to the work presented here—firstly, while location and time-period specific data were available for the collated rainfall, varying (often limited) degrees of geospatial information were present in each included study. The environmental covariates used in the multinomial-logistic regression were therefore spatially averaged over reported study area, and additionally often across multiple years due to the absence of time-period specific data. This spatio-temporal averaging may obscure relevant inter-annual variation in factors (e.g. rainfall) that affect population dynamics [23], and may contribute to some of the more limited seasonality (e.g. Clusters 2 and 4) and timing of seasonal peaks (in e.g. Clusters 1 and 3) observed. We mitigate this somewhat by extracting time-period specific rainfall data for each study but cannot preclude some role of spatio-temporal averaging in the results presented here. Another limitation is the heterogeneity in mosquito sampling methods across the studies. Studies varied in the catch-method used (landing catch, resting collections, pit collections, light traps and spray catches), as well as timing (dawn, dusk, night-time etc.) and location (typically either human-dwellings or cattlesheds) of collections. This heterogeneity may interact with mosquito traits (such as timing [65] or degree of indoor/outdoor biting [66] and host preferences [67]) that vary between species, and have implications for which species are sampled, and their comparative abundance [68]. We partially mitigate this heterogeneity by normalizing the catch data, but this incomplete accounting for differences in catch methodological characteristics might lead to biases in the presented inferences presented. There were also significant differences in the average number of mosquitoes caught between clusters, with Cluster 4 (perennial dynamics) having the lowest average catch size. While differences in catch sizes between clusters were smaller than within cluster variation (where individual study

counts ranged over several orders of magnitude and were highly overdispersed), it is possible that the lack of observed seasonality for Cluster 4 time series might be an artefact of limited sampling effort and mosquitoes caught.

Overall, our work highlights that the substantial variation in temporal dynamics across mosquito populations can be clustered into a small number of dynamical archetypes, each characterized by distinct temporal properties and associated with distinct environmental factors. In doing so, this work underscores the crucial importance of integrating both species composition and ecological structure into our understanding of the temporal profile of malaria risk and provides a generically applicable framework to better identify and understand patterns of seasonal variation in vectors relevant to public health—a crucial and operationally relevant input for optimizing the delivery of control interventions.

Data accessibility. All data collated as part of this study are available in the electronic supplementary material [69]. Analytical code used to produce these analyses (as well as a copy of the data) can be found at https://github.com/cwhittaker1000/anopheleseasonality. A static, archived version of the repository can be found via https://doi.org/10.5281/zenodo.5862952 and associated GitHub release at https://github.com/cwhittaker1000/anopheleseasonality/releases/tag/v1.0.3.

Authors' contributions. C.W.: conceptualization, data curation, formal analysis, investigation, methodology, software, visualization, writing—original draft, writing—review and editing; P.W.: conceptualization, supervision, writing—review and editing; M.S.: data curation, resources, writing—review and editing; S.P.: data curation, resources, writing—review and editing; C.M.: data curation, resources, writing—review and editing; D.J.W.: resources, writing—review and editing; M.N.: resources, writing—review and editing; P.W.G.: resources, writing—review and editing; A.K.: conceptualization, supervision, writing—review and editing; A.G.: conceptualization, funding acquisition, project administration, supervision, writing—review and editing; S.B.: conceptualization, formal analysis, investigation, methodology, resources, software, supervision, writing—review and editing.

All authors gave final approval for publication and agreed to be held accountable for the work performed therein.

Conflict of interest declaration. We declare we have no competing interests.

Funding. S.B. and A.G. both acknowledge grant support from the Bill and Melinda Gates Foundation. C.W. acknowledges funding from the MRC Centre for Global Infectious Disease Analysis (reference MR/R015600/1), jointly funded by the UK Medical Research Council (MRC) and the UK Foreign, Commonwealth & Development Office (FCDO), under the MRC/FCDO Concordat agreement and is also part of the EDCTP2 programme supported by the European Union.

Acknowledgements. C.W. is supported by a Medical Research Council Doctoral Training Partnership PhD Studentship.

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
