## [Peer Review File · Proceedings of the Royal Society B: Biological Sciences]

Review History

RSPB-2021-1386.R0 (Original submission)

Review form: Reviewer 1

Recommendation

Major revision is needed (please make suggestions in comments)

Scientific importance: Is the manuscript an original and important contribution to its field?

Excellent

General interest: Is the paper of sufficient general interest?

Good

Quality of the paper: Is the overall quality of the paper suitable?

Excellent

Is the length of the paper justified?

Yes

Should the paper be seen by a specialist statistical reviewer?

No

Do you have any concerns about statistical analyses in this paper? If so, please specify them explicitly in your report.

Yes

It is a condition of publication that authors make their supporting data, code and materials available - either as supplementary material or hosted in an external repository. Please rate, if applicable, the supporting data on the following criteria.

Is it accessible?

Yes

Is it clear?

Yes

Is it adequate?

Yes

Do you have any ethical concerns with this paper?

No

Comments to the Author

Please see attached word doc. (See Appendix A)

Review form: Reviewer 2

Recommendation

Major revision is needed (please make suggestions in comments)

Scientific importance: Is the manuscript an original and important contribution to its field?

Good

General interest: Is the paper of sufficient general interest?

Good

Quality of the paper: Is the overall quality of the paper suitable?

Good

Is the length of the paper justified?

Yes

Should the paper be seen by a specialist statistical reviewer?

Yes

Do you have any concerns about statistical analyses in this paper? If so, please specify them explicitly in your report.

No

It is a condition of publication that authors make their supporting data, code and materials available - either as supplementary material or hosted in an external repository. Please rate, if applicable, the supporting data on the following criteria.

Is it accessible?

No

Is it clear?

Yes

Is it adequate?

No

Do you have any ethical concerns with this paper?

No

Comments to the Author

Review of "The Ecological Structure of Mosquito Population Seasonal Dynamics" RSPB-2021-1386
In this paper, the authors take a remarkable number of mosquito population time series data sets from across the Indian sub-continent, and characterize them into clusters of similar dynamics, with respect to seasonal climate. This is a remarkable set of data, and I really enjoyed reading the paper, the temporal dynamics and clustering approaches are well described, and much of the code accessible.

My first suggestion is that the title reflect the contents slightly better, and include the geographic focus – perhaps add ‘across the Indian subcontinent’ or similar. This both gives a picture of the sheer breadth of area covered in the paper, AND, distinguishes this from other more regionally specific malaria papers (which the authors allude to). I think this will draw attention to the assemblage of species nicely, and allows for researchers from other regions to appreciate the difference.

My second suggestion is that since one of the major conclusions is that the population dynamics are far less governed by rainfall than expected, I would draw attention to the papers examining temperature (thermal boundaries and optima) as a driver of mosquito-borne disease transmission – in particular, (Miazgowiec et al. 2020) address this in *An. stephensi*, one of the more urban mosquitoes in the Asian study region. There are several preceding papers, examining malaria and malaria vectors, both describing the thermal curves, and mapping the results (Johnson, Lafferty, et al. 2015; Johnson, Ben-Horin, et al. 2015; Ryan et al. 2015; Mordecai et al. 2013), and a preprint of work in review (Villena et al. 2020). The authors touch on the issue that rainfall can be complicated to use as a driver, given how rainfall translates into breeding habitat, and that is another point that would be great to emphasize further, and perhaps offer some pointers to researchers in other regions (e.g. the Americas) who are approaching these questions; given the number of different habitats within the Indian subcontinent, this would be very helpful. I can see that this spatio-temporal population clustering approach would apply to many regions and multiple VBD systems.

In Figure 1, would it be feasible to add a bit more geographic information? The first panel could benefit from perhaps some context (label India, perhaps point out the administrative level being depicted by the lines, add country labels to the north, and maybe even a simple relief map as the backdrop, rather than yellow; give a scale bar). One can leverage a map to convey a LOT to readers, and given the limit on figures in articles, this is a good opportunity to do that. I would also make the internal black lines grey, and check the dot colors for colorblind contrast.

Figure 2 – if the panel layout is as in the review copy, I would put the Cluster descriptors (Monsoon, Bimodal, etc) as call out labels on the ellipses, OR, label the ellipses as Clusters 1-4, and place the labels in B as well. I really like Figure 2C, as a means to see the species communities within the clusters.

Figure 4 and the use of the predictive maps – I felt like this piece of the paper rather didn't entirely hang with the rest of it; the predictive mapping is part of another project, and the website they are on is less accessible than the data and code within this paper. The probability maps are generated using a different type of modeling, and the reliability of the output not fully described up front. I think either I would have liked to see multiple niche-generating algorithm output maps, to show the range of importance and geographic spread, or to have this piece better integrated into the main body of the paper, rather than referring the reader to a supplement elsewhere for the methods. While this is not a dealbreaker, it just sits oddly for me. In the figure, the predictive map is just presented as a given probability gradient – these are notoriously hard

to interpret, and thus I would maybe use a probability cutoff method instead (e.g. >0.5), to better represent where the reader should visually cue in on the fit.

I have no minor edits to list out here, but would happily see a revised version addressing the rearrangement of text in response to my suggestions, and updated figures.

Data and Code Access: On the GitHub repository, the labeling suggests this was a PNAS submission, and the link to 'data' in the instructions to download, does not work. This means the data are actually NOT accessible beyond the copy appended with the reviewer proof. I would recommend the authors correct this, and perhaps relabel/rename the repository (or check that the right one is linked in the paper).

Overall, I think this paper presents a useful set of methods applied to a remarkable set of data, and it will be a nice contribution to the literature, and to the toolkit for vector borne disease analyses. For this latter reason, I would really like to see a clean repository and working data access.

Johnson, Leah R., Tal Ben-Horin, Kevin D. Lafferty, Amy McNally, Erin Mordecai, Krijn P. Paaijmans, Samraat Pawar, and Sadie J. Ryan. 2015. "Understanding Uncertainty in Temperature Effects on Vector-Borne Disease: A Bayesian Approach." *Ecology* 96 (1): 203–13.

Johnson, Leah R., Kevin D. Lafferty, Amy McNally, Erin Mordecai, Krijn P. Paaijmans, Samraat Pawar, and Sadie J. Ryan. 2015. "Mapping the Distribution of Malaria: Current Approaches and Future Directions." In *Wiley Series in Probability and Statistics*, 189–209. Hoboken, NJ, USA: John Wiley & Sons, Inc.

Miazgowicz, K. L., M. S. Shocket, S. J. Ryan, O. C. Villena, R. J. Hall, J. Owen, T. Adanlawo, et al. 2020. "Age Influences the Thermal Suitability of *Plasmodium Falciparum* Transmission in the Asian Malaria Vector *Anopheles Stephensi*." *Proceedings. Biological Sciences / The Royal Society* 287 (1931): 20201093.

Mordecai, Erin A., Krijn P. Paaijmans, Leah R. Johnson, Christian Balzer, Tal Ben-Horin, Emily de Moor, Amy McNally, et al. 2013. "Optimal Temperature for Malaria Transmission Is Dramatically Lower than Previously Predicted." *Ecology Letters* 16 (1): 22–30.

Ryan, Sadie J., Amy McNally, Leah R. Johnson, Erin A. Mordecai, Tal Ben-Horin, Krijn Paaijmans, and Kevin D. Lafferty. 2015. "Mapping Physiological Suitability Limits for Malaria in Africa Under Climate Change." *Vector Borne and Zoonotic Diseases* 15 (12): 718–25.

Villena, Oswaldo C., Sadie J. Ryan, Courtney C. Murdock, and Leah R. Johnson. 2020. "Temperature Impacts the Transmission of Malaria Parasites by *Anopheles Gambiae* and *Anopheles Stephensi* Mosquitoes." Cold Spring Harbor Laboratory. <https://doi.org/10.1101/2020.07.08.194472>.

Decision letter (RSPB-2021-1386.R0)

02-Aug-2021

Dear Mr Whittaker:

I am writing to inform you that your manuscript RSPB-2021-1386 entitled "The Ecological Structure of Mosquito Population Seasonal Dynamics" has, in its current form, been rejected for publication in *Proceedings B*.

This action has been taken on the advice of referees, who have recommended that substantial revisions are necessary. With this in mind we would be happy to consider a resubmission, provided the comments of the referees are fully addressed. However please note that this is not a provisional acceptance.

The resubmission will be treated as a new manuscript. However, we will approach the same reviewers if they are available and it is deemed appropriate to do so by the Editor. Please note

that resubmissions must be submitted within six months of the date of this email. In exceptional circumstances, extensions may be possible if agreed with the Editorial Office. Manuscripts submitted after this date will be automatically rejected.

Sincerely,
 Professor Hans Heesterbeek
 mailto: proceedingsb@royalsociety.org

Associate Editor
 Board Member: 1
 Comments to Author:

We have now received two reviews of this manuscript. While the reviewers see this work as having the potential to make an important contribution to the literature, both have raised substantial issues that need to be addressed to improve the manuscript. In addition to the notes raised by the reviewers, additional attention needs to be given to making sure all of the necessary code and data are made accessible. While the code for the population dynamics part of the study is available via GitHub, as reviewer 2 notes, the data are not currently accessible. In addition, the code and data should be made available for the predictive mapping portion of the study, and this is currently missing.

Reviewer(s)' Comments to Author:
 Referee: 1
 Comments to the Author(s)
 Please see attached word doc.

Referee: 2

Comments to the Author(s)

Review of "The Ecological Structure of Mosquito Population Seasonal Dynamics" RSPB-2021-1386
 In this paper, the authors take a remarkable number of mosquito population time series data sets from across the Indian sub-continent, and characterize them into clusters of similar dynamics, with respect to seasonal climate. This is a remarkable set of data, and I really enjoyed reading the paper, the temporal dynamics and clustering approaches are well described, and much of the code accessible.

My first suggestion is that the title reflect the contents slightly better, and include the geographic focus - perhaps add 'across the Indian subcontinent' or similar. This both gives a picture of the sheer breadth of area covered in the paper, AND, distinguishes this from other more regionally specific malaria papers (which the authors allude to). I think this will draw attention to the

assemblage of species nicely, and allows for researchers from other regions to appreciate the difference.

My second suggestion is that since one of the major conclusions is that the population dynamics are far less governed by rainfall than expected, I would draw attention to the papers examining temperature (thermal boundaries and optima) as a driver of mosquito-borne disease transmission – in particular, (Miazgowiec et al. 2020) address this in *An. stephensi*, one of the more urban mosquitoes in the Asian study region. There are several preceding papers, examining malaria and malaria vectors, both describing the thermal curves, and mapping the results (Johnson, Lafferty, et al. 2015; Johnson, Ben-Horin, et al. 2015; Ryan et al. 2015; Mordecai et al. 2013), and a preprint of work in review (Villena et al. 2020). The authors touch on the issue that rainfall can be complicated to use as a driver, given how rainfall translates into breeding habitat, and that is another point that would be great to emphasize further, and perhaps offer some pointers to researchers in other regions (e.g. the Americas) who are approaching these questions; given the number of different habitats within the Indian subcontinent, this would be very helpful. I can see that this spatio-temporal population clustering approach would apply to many regions and multiple VBD systems.

In Figure 1, would it be feasible to add a bit more geographic information? The first panel could benefit from perhaps some context (label India, perhaps point out the administrative level being depicted by the lines, add country labels to the north, and maybe even a simple relief map as the backdrop, rather than yellow; give a scale bar). One can leverage a map to convey a LOT to readers, and given the limit on figures in articles, this is a good opportunity to do that. I would also make the internal black lines grey, and check the dot colors for colorblind contrast.

Figure 2 – if the panel layout is as in the review copy, I would put the Cluster descriptors (Monsoon, Bimodal, etc) as call out labels on the ellipses, OR, label the ellipses as Clusters 1-4, and place the labels in B as well. I really like Figure 2C, as a means to see the species communities within the clusters.

Figure 4 and the use of the predictive maps – I felt like this piece of the paper rather didn't entirely hang with the rest of it; the predictive mapping is part of another project, and the website they are on is less accessible than the data and code within this paper. The probability maps are generated using a different type of modeling, and the reliability of the output not fully described up front. I think either I would have liked to see multiple niche-generating algorithm output maps, to show the range of importance and geographic spread, or to have this piece better integrated into the main body of the paper, rather than referring the reader to a supplement elsewhere for the methods. While this is not a dealbreaker, it just sits oddly for me. In the figure, the predictive map is just presented as a given probability gradient – these are notoriously hard to interpret, and thus I would maybe use a probability cutoff method instead (e.g. >0.5), to better represent where the reader should visually cue in on the fit.

I have no minor edits to list out here, but would happily see a revised version addressing the rearrangement of text in response to my suggestions, and updated figures.

Data and Code Access: On the GitHub repository, the labeling suggests this was a PNAS submission, and the link to 'data' in the instructions to download, does not work. This means the data are actually NOT accessible beyond the copy appended with the reviewer proof. I would recommend the authors correct this, and perhaps relabel/rename the repository (or check that the right one is linked in the paper).

Overall, I think this paper presents a useful set of methods applied to a remarkable set of data, and it will be a nice contribution to the literature, and to the toolkit for vector borne disease analyses. For this latter reason, I would really like to see a clean repository and working data access.

Johnson, Leah R., Tal Ben-Horin, Kevin D. Lafferty, Amy McNally, Erin Mordecai, Krijn P. Paaijmans, Samraat Pawar, and Sadie J. Ryan. 2015. "Understanding Uncertainty in Temperature Effects on Vector-Borne Disease: A Bayesian Approach." *Ecology* 96 (1): 203–13.

Johnson, Leah R., Kevin D. Lafferty, Amy McNally, Erin Mordecai, Krijn P. Paaijmans, Samraat Pawar, and Sadie J. Ryan. 2015. "Mapping the Distribution of Malaria: Current Approaches and Future Directions." In *Wiley Series in Probability and Statistics*, 189–209. Hoboken, NJ, USA: John Wiley & Sons, Inc.

Miazgowicz, K. L., M. S. Shocket, S. J. Ryan, O. C. Villena, R. J. Hall, J. Owen, T. Adanlawo, et al. 2020. "Age Influences the Thermal Suitability of Plasmodium Falciparum Transmission in the Asian Malaria Vector Anopheles Stephensi." *Proceedings. Biological Sciences / The Royal Society* 287 (1931): 20201093.

Mordecai, Erin A., Krijn P. Paaijmans, Leah R. Johnson, Christian Balzer, Tal Ben-Horin, Emily de Moor, Amy McNally, et al. 2013. "Optimal Temperature for Malaria Transmission Is Dramatically Lower than Previously Predicted." *Ecology Letters* 16 (1): 22–30.

Ryan, Sadie J., Amy McNally, Leah R. Johnson, Erin A. Mordecai, Tal Ben-Horin, Krijn Paaijmans, and Kevin D. Lafferty. 2015. "Mapping Physiological Suitability Limits for Malaria in Africa Under Climate Change." *Vector Borne and Zoonotic Diseases* 15 (12): 718–25.

Villena, Oswaldo C., Sadie J. Ryan, Courtney C. Murdock, and Leah R. Johnson. 2020. "Temperature Impacts the Transmission of Malaria Parasites by Anopheles Gambiae and Anopheles Stephensi Mosquitoes." Cold Spring Harbor Laboratory. <https://doi.org/10.1101/2020.07.08.194472>.

Author's Response to Decision Letter for (RSPB-2021-1386.R0)

See Appendices B & C.

RSPB-2022-0089.R0

Review form: Reviewer 1

Recommendation

Accept with minor revision (please list in comments)

Scientific importance: Is the manuscript an original and important contribution to its field?

Excellent

General interest: Is the paper of sufficient general interest?

Good

Quality of the paper: Is the overall quality of the paper suitable?

Excellent

Is the length of the paper justified?

Yes

Should the paper be seen by a specialist statistical reviewer?

No

Do you have any concerns about statistical analyses in this paper? If so, please specify them explicitly in your report.

No

It is a condition of publication that authors make their supporting data, code and materials available - either as supplementary material or hosted in an external repository. Please rate, if applicable, the supporting data on the following criteria.

Is it accessible?

Yes

Is it clear?

Yes

Is it adequate?

Yes

Do you have any ethical concerns with this paper?

No

Comments to the Author

Please see attached minor comments. (See Appendix D)

Review form: Reviewer 2

Recommendation

Accept with minor revision (please list in comments)

Scientific importance: Is the manuscript an original and important contribution to its field?

Good

General interest: Is the paper of sufficient general interest?

Good

Quality of the paper: Is the overall quality of the paper suitable?

Good

Is the length of the paper justified?

Yes

Should the paper be seen by a specialist statistical reviewer?

No

Do you have any concerns about statistical analyses in this paper? If so, please specify them explicitly in your report.

No

It is a condition of publication that authors make their supporting data, code and materials available - either as supplementary material or hosted in an external repository. Please rate, if applicable, the supporting data on the following criteria.

Is it accessible?

Yes

Is it clear?

Yes

Is it adequate?

Yes

Do you have any ethical concerns with this paper?

No

Comments to the Author

I appreciate the authors' attention to the reviewer comments, and the responses are largely adequate. I really appreciate the clean GitHub repo and attention to detail - it might be useful to connect to a service such as Dryad for storage of larger files, so that the project doesn't deprecate and require future researchers to contact the authors about shapefiles, e.g. This is just a suggestion for making the project a useful package, rather than a necessary fix at this stage. If the intent is for future users to use the tools and be able to see the full implementation, being able to locate the data easily is part of that practice, but I also appreciate it's beyond the scope of journal obligation.

I now really like Figure 2, and am glad that my minor suggestions give it a bit more cohesion, and I think it really sings.

I would like to see a justification for the probability cutoff for the spatial mapping component - the word thresholded is misspelled in the figure legend (minor correction). I wasn't sure where the choice of 0.67 comes from, and I would like to know that 2/3 majority has meaning in the context of predicting presence. Perhaps the authors could provide evidence from an external validation exercise, or similar.

Lastly, I still think the title should be clear that this is India, or, if the intent is, as the authors response suggests, to introduce a novel set of tools, make the title more indicative of that aspect instead. I want to emphasize that it is very useful to signpost that these are novel methods and approaches, so that future researchers will apply them and be able to challenge them with mosquito (and other species') population trends from other locations.

In summary, I am very keen on this paper, it is still a pleasure to read, and I look forward to seeing the final version.

Decision letter (RSPB-2022-0089.R0)

21-Feb-2022

Dear Mr Whittaker:

Your manuscript has now been peer reviewed and the reviews have been assessed by an Associate Editor. The reviewers' comments (not including confidential comments to the Editor) and the comments from the Associate Editor are included at the end of this email for your reference. As you will see, the reviewers are very positive about the revision but have also raised some issues that we would like you to address.

When submitting your revision please upload a file under "Response to Referees" in the "File Upload" section. This should document, point by point, how you have responded to the

reviewers' and Editors' comments, and the adjustments you have made to the manuscript. We require a copy of the manuscript with revisions made since the previous version marked as 'tracked changes' to be included in the 'response to referees' document.

Research ethics:

Use of animals and field studies:

It is a condition of publication that you make available the data and research materials supporting the results in the article (<https://royalsociety.org/journals/authors/author-guidelines/#data>). Datasets should be deposited in an appropriate publicly available repository and details of the associated accession number, link or DOI to the datasets must be included in the Data Accessibility section of the article (<https://royalsociety.org/journals/ethics-policies/data-sharing-mining/>). Reference(s) to datasets should also be included in the reference list of the article with DOIs (where available).

Online supplementary material will also carry the title and description provided during submission, so please ensure these are accurate and informative. Note that the Royal Society will not edit or typeset supplementary material and it will be hosted as provided. Please ensure that

the supplementary material includes the paper details (authors, title, journal name, article DOI). Your article DOI will be 10.1098/rspb.[paper ID in form xxxx.xxxx e.g. 10.1098/rspb.2016.0049].

Please submit a copy of your revised paper within three weeks. If we do not hear from you within this time your manuscript will be rejected. If you are unable to meet this deadline please let us know as soon as possible, as we may be able to grant a short extension.

Best wishes,
Professor Hans Heesterbeek
mailto: proceedingsb@royalsociety.org

Reviewer(s)' Comments to Author:
Referee: 1
Comments to the Author(s).
Please see attached minor comments.

Referee: 2
Comments to the Author(s).

I appreciate the authors' attention to the reviewer comments, and the responses are largely adequate. I really appreciate the clean GitHub repo and attention to detail - it might be useful to connect to a service such as Dryad for storage of larger files, so that the project doesn't deprecate and require future researchers to contact the authors about shapefiles, e.g. This is just a suggestion for making the project a useful package, rather than a necessary fix at this stage. If the intent is for future users to use the tools and be able to see the full implementation, being able to locate the data easily is part of that practice, but I also appreciate it's beyond the scope of journal obligation.

I now really like Figure 2, and am glad that my minor suggestions give it a bit more cohesion, and I think it really sings.

I would like to see a justification for the probability cutoff for the spatial mapping component - the word thresholded is misspelled in the figure legend (minor correction). I wasn't sure where the choice of 0.67 comes from, and I would like to know that 2/3 majority has meaning in the context of predicting presence. Perhaps the authors could provide evidence from an external validation exercise, or similar.

Lastly, I still think the title should be clear that this is India, or, if the intent is, as the authors response suggests, to introduce a novel set of tools, make the title more indicative of that aspect instead. I want to emphasize that it is very useful to signpost that these are novel methods and approaches, so that future researchers will apply them and be able to challenge them with mosquito (and other species') population trends from other locations.

In summary, I am very keen on this paper, it is still a pleasure to read, and I look forward to seeing the final version.

Author's Response to Decision Letter for (RSPB-2022-0089.R0)

See Appendix E.

Decision letter (RSPB-2022-0089.R1)

01-Mar-2022

Dear Mr Whittaker

I am pleased to inform you that your manuscript entitled "A Novel Statistical Framework for Exploring the Population Dynamics and Seasonality of Mosquito Populations" has been accepted for publication in Proceedings B.

Data Accessibility section

Open Access

You are invited to opt for Open Access, making your freely available to all as soon as it is ready for publication under a CCBY licence. Our article processing charge for Open Access is £1700. Corresponding authors from member institutions (<http://royalsocietypublishing.org/site/librarians/allmembers.xhtml>) receive a 25% discount to these charges. For more information please visit <http://royalsocietypublishing.org/open-access>.

Paper charges

Sincerely,

Professor Hans Heesterbeek

Appendix A

This paper presents a meta-analysis of longitudinal catch data for Anopheline mosquitoes in India obtained through systematic review of the literature. The study uses a framework to statistically characterise the structure of the time-series data and shows that there are four general seasonal patterns each of which driven by different environmental factors. Overall, it was a pleasure to read this piece of work. The study is novel, and I think the approach, which combines both temporal and geospatial analyses, will be of interest to entomologists and more generally ecologists. While I can't comment in detail on the statistical analyses, they seemed appropriate to address the aims of the study and I understood the approach sufficiently to be able to interpret the results and place into a broader context. While this work focuses on Anopheline mosquitoes the approach and conclusions are likely an important consideration for all vector-borne disease systems. If the queries below are addressed, I would consider the scientific approach and manuscript to be excellent, of general interest and of publishable quality for Proc. B.

There was no line numbering in the draft manuscript, so I've quoted text in italics where appropriate or referred to section and paragraph number.

Major comments

These should be relatively easy to fix but, as is, they stick out as problematic when reading.

1. Reference to '*often assumed positive relationship with rainfall*'

Abstract: '*Many mosquito populations lacked the often-assumed positive relationship with rainfall...*'

Background: '*Rainfall is frequently considered a key determinant of mosquito temporal dynamics*' – this is probably fair enough as stated but need references to support and by whom.

Results: '*...many of the collated time-series lacking the close, positive correlation with rainfall typically assumed for mosquito populations*'

Discussion: '*Many of the populations studied here lacked the frequently assumed positive relationship with rainfall...*'

These statements are not supported by reference to the literature, nor is it stated who has made this assumption. I think it widely accepted by both field biologists and modelers that rainfall is often an important driver of mosquito population dynamics, but this does not equate to assuming a positive correlation in time – the situation is, of course, far more complex. I also do not think you need these statements to justify your study. I would suggest removing these statements or otherwise backing up with references from the published literature.

2. Role of temperature appears to be a focus – should it be?

a) Abstract: '*Highlighting the role of temperature...in shaping the dynamics...*'

Background: '*the influence of other factors, such as temperature remains similarly unclear...influence of temperature regimen on mosquito population dynamics largely unexplored in other ecological settings.*'

Discussion: '*rainfall is frequently considered a key driver of mosquito population dynamics but the role of temperature in shaping mosquito population dynamics is increasingly being recognised (39).*'

While temperature likely plays a role in many situations, consider here giving a more balanced argument for a broader consideration of 'ecological context' including temperature, hydrology, irrigation and artificial larval habitats, rather than mentioning temperature alone.

Ref: 39 Sahu et. al. (2017) I fail to see how this reference supports the statement that the role of temperature in shaping mosquito population dynamics is increasingly being recognised? There is brief mention of differences in abundance between seasons in this reference, but it doesn't appear to relate directly to temperature, nor the role of temperature relative to rainfall, unless I've missed something.

b) Discussion, third paragraph: *'We identified a significant impact of temperature on population dynamics, with temperature seasonality strongly positively associated with the highly seasonal, monsoon peaking seasonal dynamics...together these results suggest a role for both in shaping annual patterns...'*

Can you tease apart the relative contribution of both rainfall (or other individual factors that covary) and temperature here? Does temperature contribute anything once rainfall has been accounted for? I am worried that this is just because in many circumstances temperature and rainfall will not vary independently? If I have misunderstood, perhaps clarification here would be useful for other readers too. It's not clear to me from your analyses that temperature explains any more after rainfall is accounted for or if your analyses can show that.

3. There needs to be a limitations section in the Discussion

Please add a few sentences to discuss variation in catch methods in the data set and averaging climate data over space and time and refer back to any extra analyses you might be able to do (see below and point 4) to reassure readers that these limitations do not affect your findings. All this important detail is currently hidden a little in the supplementary material and I think it important to be a bit more explicit in the main manuscript (I've made specific suggestions below).

a) Environmental variables: great that you can find these four patterns using averaged climate data over time and at a 5km x 5km resolution. Can you say despite the fact you didn't have contemporary rainfall etc data to match to the exact times and locations from which times-series were obtained, the analysis still shows general trends? Or something to that effect. I think it important to state in the method section the resolution, period over which data were aggregated over and source of the imagery used to estimate environmental variables. This is particularly important for climate data, which is likely variable between years and locations. The paper referenced in the introduction; Mendis et al. (2000) has a good example of interannual variation which would be lost in averaging over many years. I think you need to acknowledge this may possibly have contributed to those time-series in Cluster 2 and 4? Great if you can provide evidence why this is not the case.

b) Catch methods: I would state in the methods section the number of different trapping methods used – it need just be a sentence to summarise and then this also needs to be acknowledged in a limitations section in the Discussion. While you normalise the data so that comparisons are relative, this may not account for all biases in differences in trap catches resulting from their different attractiveness for females at different reproductive stages and behaviour, including dispersal, at a given time & influenced by local climate.

Could you compare the time-series that cluster in either the dry or monsoon peak with those that are perennial and see if there is a correlation with raw catch size? I worry that the lack of seasonality may be due to low catches year-round which could be due to catch method and

hence introduce a subtle bias into your analyses and influence the reliability of the conclusions you make from this. Happy to be persuaded otherwise.

4. Sample size for each species and results

In fourth paragraph of results and with reference to Fig. 3A: *'in contrast to An. culicifacies s.l. and An. subpictus (which clustered together and showed a strong positive assoc. with C1 and -ve with C3), binary indicator for An. dirus displayed weak associations with all clusters, including C1.'*

Looking at Fig. 3A – Fluviatilis appears to show the strongest correlations (positive for C3 and negative for C1), followed by culicifacies and subpictus, then annularis, stephensi, dirus then minimus.

If we then also look at time-series sample size, fluviatilis – 60, culicifacies – 85, subpictus – 38, annularis – 39, stephensi – 27, dirus – 11, minimus – 12. Is it possible that the lack of association to a specific cluster is an artefact of small sample size?

In the least this needs to be acknowledged and addressed in the Discussion. Could you sub-sample the larger datasets randomly to simulate what would happen if you only had say 10 time-series for fluviatilis for example?

5. In the Background, references 22 and 23 are used to argue that *'...An. funestus and An. annularis populations frequently lack marked seasonal fluctuations in population abundance (22,23 10,24,25). This brings into question how generalisable relationships between rainfall and mosquito population dynamics are'*.

I have only looked at references 22 and 23 with respect to *An. funestus*, but these don't support this statement. There is seasonality in *An. funestus* abundance catches as shown in Fig. 1 of Cohuet *et al* (2004) and this generally appears to follow rainfall trends. While >100 individuals were caught at all times of year, there is still seasonality, and I would hypothesise that the perennial transmission was in part due to a relatively short dry season and plenty of rain at other times of year (compare monthly rainfall in this paper with that of Mendis *et al* 2000). For Mendis *et al.* (2000), as shown in their Fig. 2, relatively few *An. funestus* were caught during the entire study and whether sampling was sufficient to be able to detect seasonality is questionable. Here monthly rainfall was usually <50mm and only >150mm twice in c. 1.5yrs. In the Cohuet study rainfall was frequently >100mm and >150mm 8 times. Hence, here the question is being asked whether we can generalise about relationships between rainfall and mosquito population dynamics but the evidence you provide, at least for *An. funestus*, is supportive of a positive relationship between *An. funestus* catches and rainfall.

I would suggest you either provide references that support this statement or change this statement. It might also help if you can be more quantitative in your statements. For example, here *'lacked marked seasonal fluctuations'* could be changed to consider relative differences between species from one or two studies?

Minor comments

Abstract

6. *'Diverse dynamics can be clustered into 'dynamical archetypes', each characterised by distinct temporal properties and driven largely by a unique set of environmental factors'*

Would suggest remove 'driven' and change to stating 'correlate with' as you don't show mechanistic relationships between the seasonal dynamics and potential explanatory variables. Would also state that they were clustered into 'four 'dynamical archetypes''. I think a nice finding of the study were that there were just these four clear groups/ patterns.

7. *'These results highlight that a complex interplay of factors, rather than rainfall alone shape the timing and extent of mosquito population seasonality'*

Could you alter slightly to end abstract on more interesting note?! From my perspective, this conclusion is a bit obvious and not terribly exciting. Either potentially something about the approach and application to other taxa (if it's novel) or the importance of considering 'archetypes' – what would that benefit disease ecology research and vector control?

Background

8. Although mentioned in the Discussion, I would also briefly summarise here the different larval habitats associated with the different Anophelines if you can afford the word count.
9. Following from comment 1 on text which provides motivation for the study, you could consider stating that while many entomological studies have been carried out, they are rarely brought together to be able to make generalisations and to compare and contrast among vector species/ contexts – this was the primary motivation for my interest in the study.

Results

10. Can you be more quantitative throughout the text? E.g. 1st paragraph a) '**substantial variation** in temporal dynamics was observed between different species complexes with **many** of the collated time-series lacking the close, positive correlation with rainfall typically assumed for mosquito populations'; b) 'despite highly seasonal patterns of rainfall, **a number of** time-series belonging to *An. annularis* s.l. demonstrated perennial patterns of abundance'; c) **A number of environmental covariates** also demonstrated cluster-specific associations...'

You could delete the sentence a) in the first paragraph which would deal with Comment 1 and make this section more succinct. You then follow on to describe this statement in better detail anyway. Also suggest deleting sentence c) as you can go straight into saying this explicitly.

11. For b) in 10 how many 2? 10? 50? This detail is important – for these studies when were they carried out and have you checked if in the manuscript or elsewhere there are local rainfall data available which may explain the perennial pattern? E.g., back to

the averaging question in major comments – it may be that your averaged rainfall is not of sufficient resolution – perhaps these were particularly dry periods.

12. In third paragraph '*...indicating different sibling species within a complex display distinct temporal dynamics or that mosquito populations belonging to the species complex are able to adopt a diverse array of temporal dynamics depending on the particular ecological setting*' suggest move to discussion and also change word 'adopt' as make it sound intentional rather than an emergent phenomenon. Similarly, '*This suggests that the ecological features of locations of *An. dirus* had been sampled in rather than intrinsic to species-complex itself were primary driver of observed dynamics*' move to Discussion.

Figures

Fig 1 – are time series over many different years? While this can be found in the supplementary material and may be mentioned in the methods, would be good to state the years over which data were collected in the legend.

Fig 2 – y axis units different from fig 1, assume fig 1 need to multiply by 100

Fig 3 – What is CHIRPS?

Discussion

13. Third paragraph: '*...our results also support a significant role for the environment in shaping mosquito population dynamics...*'

Be more specific or delete as of course the environment is important in shaping mosquito population dynamics.

14. Third paragraph: '*Many of the populations studied here lacked the frequently assumed positive relationship with rainfall and instead displayed patterns of abundance that were only weakly or even negatively correlated. Rainfall is frequently considered a key driver of mosquito population dynamics...*'

'Many' – be quantitative here – at least more specific about which groups? Unless I've misunderstood your analyses don't account for temporal delays in the effect of rainfall? – e.g. a peak in mosquito abundance often likely doesn't coincide well with the peak in rainfall due to dynamics. I think this should be acknowledged here.

Appendix B

MRC Centre for Global Infectious Disease Analysis

Department of Infectious Disease Epidemiology

Imperial College London

St Mary's Campus, Norfolk Place

London W2 1PG

United Kingdom

charles.whittaker16@imperial.ac.uk

16th January 2022

Charles Whittaker

Research Postgraduate

Dear Editors,

Please find enclosed our resubmitted manuscript titled "*The Ecological Structure of Mosquito Population Seasonal Dynamics*" (reference number RSPB-2021-1386) for your consideration.

In response to the comments offered by the two reviewers, we have revised the manuscript considerably, including a shift in focus to highlight the generic applicability of the presented framework to vectors of disease more generally, a number of additional analyses that highlight the robustness of the inferences made, and a more detailed consideration of this study's limitations in the discussion section.

We feel these additions have materially improved the manuscript and so thank the reviewers for their immensely helpful suggestions. Attached alongside the revised manuscript is a complete, point-by-point description of how we have responded to the reviewer comments, and the adjustments we have made to the manuscript as a result.

The work presented here holds relevance for those concerned with the epidemiology of vector-borne diseases such as malaria, particularly those actively working on development and implementation of vector control interventions. It also provides insight for those concerned with the fundamental ecology of mosquitoes and the factors driving seasonal dynamics in population abundance. This will, we hope, make this study of interest to the wide readership of *Proceedings of the Royal Society B: Biological Sciences*.

This revised manuscript describes original work and has not been published. None of its components are under consideration in any other peer-reviewed journal. All authors listed approved the manuscript, this submission and have all contributed sufficiently to the project.

Thank you for your consideration and if you require anything else at this stage, please do not hesitate to get in touch.

Sincerely,

Charles Whittaker

Corresponding Author

Appendix C

Reviewer 1

This paper presents a meta-analysis of longitudinal catch data for Anopheline mosquitoes in India obtained through systematic review of the literature. The study uses a framework to statistically characterise the structure of the time-series data and shows that there are four general seasonal patterns each of which driven by different environmental factors. Overall, it was a pleasure to read this piece of work. The study is novel, and I think the approach, which combines both temporal and geospatial analyses, will be of interest to entomologists and more generally ecologists. While I can't comment in detail on the statistical analyses, they seemed appropriate to address the aims of the study and I understood the approach sufficiently to be able to interpret the results and place into a broader context. While this work focuses on Anopheline mosquitoes the approach and conclusions are likely an important consideration for all vector-borne disease systems. If the queries below are addressed, I would consider the scientific approach and manuscript to be excellent, of general interest and of publishable quality for Proc. B.

We thank the Reviewer immensely for their detailed comments and suggestions, which we feel have significantly improved the manuscript we now submit for reconsideration. Below is a point-by-point response to each of the Reviewer's comments, detailing how and in what way we have sought to address their comments/suggestions.

There was no line numbering in the draft manuscript, so I've quoted text in italics where appropriate or referred to section and paragraph number.

Major comments

These should be relatively easy to fix but, as is, they stick out as problematic when reading.

1. Reference to '*often assumed positive relationship with rainfall*'

Abstract: '*Many mosquito populations lacked the often-assumed positive relationship with rainfall...*'

Background: '*Rainfall is frequently considered a key determinant of mosquito temporal dynamics*' – this is probably fair enough as stated but need references to support and by whom.

Results: '*...many of the collated time-series lacking the close, positive correlation with rainfall typically assumed for mosquito populations*'

Discussion: '*Many of the populations studied here lacked the frequently assumed positive relationship with rainfall...*'

These statements are not supported by reference to the literature, nor is it stated who has made this assumption. I think it widely accepted by both field biologists and modelers that rainfall is often an important driver of mosquito population dynamics, but this does not equate to assuming a positive correlation in time – the situation is, of course, far more complex. I also do not think you need these statements to justify your study. I would suggest removing these statements or otherwise backing up with references from the published literature.

We thank the reviewer for these comments – we have removed these statements from the relevant sections of the manuscript and, per their further comments below, added further text to the Introduction to reduce the focus exclusively on rainfall and temperature and note the importance of a wider array of environmental factors.

1. Role of temperature appears to be a focus – should it be?

a) Abstract: '*Highlighting the role of temperature...in shaping the dynamics...*'

Background: '*the influence of other factors, such as temperature remains similarly unclear...influence of temperature regimen on mosquito population dynamics largely unexplored in other ecological settings.*'

Discussion: '*rainfall is frequently considered a key driver of mosquito population dynamics but the role of temperature in shaping mosquito population dynamics is increasingly being recognised (39).*'

While temperature likely plays a role in many situations, consider here giving a more balanced argument for a broader consideration of ‘ecological context’ including temperature, hydrology, irrigation and artificial larval habitats, rather than mentioning temperature alone.

We entirely agree with the Reviewer that a myriad of different ecological factors shape and influence the dynamics of *Anopheline* mosquito populations (this and the fact that different ecological factors contribute to distinct types of dynamics is a key insight of this work). We appreciate the Reviewer’s suggestion to widen the focus and in doing so give more consideration to a wider variety of ecological factors. However, we note that this request is somewhat at odds with those of Reviewer 2, who requested that we “*draw attention to the papers examining temperature (thermal boundaries and optima) as a driver of mosquito-borne disease transmission*”. We also note in the Discussion (Lines 362-376) a paragraph dedicated to exploring the role of factors other than temperature (specifically the hydrological environment and land-use structure, which is related to what the Reviewer requests).

We have added text to balance the requests of both Reviewers where possible, and so have not removed the existing text/references that explore the role of temperature. Instead we have added further text to the above paragraph in the Discussion to try and further emphasise that the influences of mosquito population dynamics are not limited to rainfall and temperature, and that instead these dynamics are a function of the wider environmental context as Reviewer 1 rightly highlights. We have also modified the Abstract to assign less prominence to the discussion of temperature in relation to the other environmental factors mentioned, and added text to the Introduction on other ecological factors and their influence on dynamics to balance the current focus on temperature (as well as address other comments from Reviewer 1 about the final sentence). That section now reads:

Abstract (Lines 25-29): “*Our results highlight that a range of environmental factors including rainfall, temperature, proximity to static water bodies and patterns of land use (particularly urbanicity) shape the dynamics and seasonality of mosquito populations; and provide a generically applicable framework to better identify and understand patterns of seasonal variation in vectors relevant to public health.*”

Introduction (Lines 68-72): “*Previous work has also suggested a potential role for numerous other ecological factors in shaping mosquito population dynamics, including land-use (such as irrigative practices³⁰ or structure of the built-environment in urban settings³¹) or the local hydrological environment and presence of long-lived water bodies^{32,33} (which potentially provide opportunities for breeding year-round).*”

Ref: 39 Sahu et. al. (2017) I fail to see how this reference supports the statement that the role of temperature in shaping mosquito population dynamics is increasingly being recognised? There is brief mention of differences in abundance between seasons in this reference, but it doesn’t appear to relate directly to temperature, nor the role of temperature relative to rainfall, unless I’ve missed something.

We thank the reviewer for flagging the issue surrounding Ref 39 (Sahu et al) – its inclusion was erroneous. The intended reference was:

- Beck-Johnson, L. M. et al. *The importance of temperature fluctuations in understanding mosquito population dynamics and malaria risk. R. Soc. Open Sci.* **4**, (2017).

which was Ref 28 in the submitted manuscript. We have corrected this mistake and also added the following reference:

- Mordecai, E. A. et al. *Thermal biology of mosquito-borne disease. Ecology Letters*, **22**, (2019).

which provides an even more recent and comprehensive overview of the ways in which temperature can influence mosquito life traits (and hence their population dynamics).

b) Discussion, third paragraph: ‘We identified a significant impact of temperature on

population dynamics, with temperature seasonality strongly positively associated with the highly seasonal, monsoon peaking seasonal dynamics...together these results suggest a role for both in shaping annual patterns...'

Can you tease apart the relative contribution of both rainfall (or other individual factors that covary) and temperature here? Does temperature contribute anything once rainfall has been accounted for? I am worried that this is just because in many circumstances temperature and rainfall will not vary independently? If I have misunderstood, perhaps clarification here would be useful for other readers too. It's not clear to me from your analyses that temperature explains any more after rainfall is accounted for or if your analyses can show that.

Within this framework, we do in fact have the ability to tease apart the relative associations of both rainfall and temperature – this arises from the fact that we include multiple ecological covariates within the same multinomial logistic regression model, and so the inferred association between cluster membership and a particular environmental covariate (e.g. Annual Mean Temperature) is the association after adjusting and controlling for all other environmental covariates (e.g. Annual Rain) included in the model (as with any other type of regression involving multiple dependent variables). This means that the reported associations for temperature-related covariates (“Annual Mean Temperature”, “Temperature Seasonality” and “Mean Temperature in the Driest Quarter”) and rainfall-related covariates (“Annual Rain”, “Rainfall Seasonality” and “Minimum Rain in the Coldest Quarter”) are all after controlling for the effect of all other covariates in the model.

The Reviewer rightly notes that multicollinearity between the ecological covariates included in the model might lead to spuriously inferred associations. We mitigate the potential impact of multicollinearity in two keyways. Firstly, we utilised only a subset (25) of the full suite (66) of environmental covariates available, with the covariates selected in order to reduce the degree of correlation between them (detailed in the Supplementary Information, section “Environmental Covariate Assembly”) – the distribution of correlation coefficients for each pair of included environmental covariates are shown in the newly included Supplementary Figure 8, and highlight the only limited degree of multi-collinearity amongst the ecological covariates included in the model. Secondly, within our multinomial logistic regression framework, we utilise an L2 (ridge) penalty. This widely used statistical technique (also known as regularisation) stabilises inferred associations between dependent and independent variables in the presence of multi-collinearity.

However, we agree with the Reviewer that these aspects of our analyses are insufficiently described. We have therefore added the following text

Supplementary Information (Lines 146-150): *“The association between each of these environmental covariates and membership of each dynamical archetype/cluster was then assessed using a Bayesian, regularised multinomial logistic regression-based framework (described in further detail below), that analyses the relationship linking each covariate to cluster/archetype membership whilst controlling for all other covariates included in the model.”*

Supplementary Information (Lines 362-365): *“Thus, within this framework (and multiple regression more generally) the inferred association between the dependent variable and each independent variable implicitly controls for all other dependent variables (environmental covariates in this case) included in the model.”*

2. There needs to be a limitations section in the Discussion

Please add a few sentences to discuss variation in catch methods in the data set and averaging climate data over space and time and refer back to any extra analyses you might be able to do (see below and point 4) to reassure readers that these limitations do not affect your findings. All this important detail is currently hidden a little in the supplementary material and I think it important to be a bit more explicit in the main manuscript (I've made specific suggestions below).

We thank the Reviewer for these greatly helpful comments – we've carried out a number of new analyses (detailed under each point below) and adapted the main text accordingly.

a) Environmental variables: great that you can find these four patterns using averaged

climate data over time and at a 5km x 5km resolution. Can you say despite the fact you didn't have contemporary rainfall etc data to match to the exact times and locations from which times-series were obtained, the analysis still shows general trends? Or something to that effect. I think it important to state in the method section the resolution, period over which data were aggregated over and source of the imagery used to estimate environmental variables. This is particularly important for climate data, which is likely variable between years and locations. The paper referenced in the introduction; Mendis et al. (2000) has a good example of interannual variation which would be lost in averaging over many years. I think you need to acknowledge this may possibly have contributed to those time-series in Cluster 2 and 4? Great if you can provide evidence why this is not the case.

We thank the Reviewer for their comment and apologise that the manuscript was unclear – but we do in fact utilise contemporary rainfall data matching the exact times (unless the study was prior to 1981 in which case we used rainfall data from the year 1981) and location (aggregated to the level of spatial granularity made possible by details on the sampling location available in the reference). We use the CHIRPS (*Climate Hazards Group InfraRed Precipitation with Station*, see <https://www.nature.com/articles/sdata201566> for further information) rainfall data which has daily rainfall available from the year 1981 onwards at a 0.05° resolution (approx. 5km by 5km). Thus, subject to some limited temporal and spatial constraints (e.g. the extent to which the study was geolocatable based on information provided in the manuscript and whether or not the study was conducted prior to 1981), we **were** able to utilise contemporaneous rainfall data for each study.

We agree this aspect of our analyses is less than clear, however. To address this confusion, we have added a figure to Figure 2 (new Figure 2B) which summarises the cross-correlation coefficient between number of mosquitoes caught and rainfall for each study, disaggregated by cluster, and added text to the legend highlighting the source and contemporaneous nature of the rainfall data used to generate the figure. These results support our findings that the population dynamics of each archetype represent genuinely differences in their relation to rainfall and are not an artefact of inter-annual variation in the dynamics of rainfall. We have also added the following text to the Methods and Results:

Methods (Lines 136-139): *For each of the 117 study locations we extracted a suite of environmental variables derived from satellite data that together describe the location's ecological structure. These include time-period and location specific rainfall data from The Climate Hazards Group Infrared Precipitation With Stations (CHIRPS) dataset²¹ ...*

Figure 2 Legend (Lines 269-271): *“Rainfall data is specific to study location and time-period and was extracted from the The Climate Hazards Group Infrared Precipitation With Stations (CHIRPS) dataset²¹.”*

Results (Lines 183-186): *“The distinct patterns displayed by each group were not due to differences in the timing and extent of rainfall across India –we collated location and time-period specific rainfall data for each study (collated from the CHIRPS dataset³⁷) and calculated the cross-correlation between mosquito density and rainfall.”*

This is in addition to Supplementary Figure 4, present in the previous manuscript draft, that plots location and time-period specific patterns of rainfall for each of the studies considered in this work, and which shows that patterns of rainfall across the locations and study time-periods being considered are largely similar. We have also modified and added the following text in the Supplementary Information:

Supplementary Information (Lines 165-174): *“In addition to the environmental covariates detailed above, for each of the 117 geolocated study locations, daily rainfall estimates specific to the location and time-period the study was conducted in were also collated. These data were taken from “The Climate Hazards Group Infrared Precipitation With Stations” (CHIRPS) dataset²¹ and were subsequently aggregated up to the same temporal resolution as the mosquito catch data (i.e. monthly). Data from the CHIRPS dataset is only available from the year 1981, and so for locations where the sampling date predated this, daily rainfall data was extracted for the year 1981, and assumed to be representative of past rainfall. These rainfall data were used to calculate the cross-correlation coefficient between mosquito catches and rainfall. See Supplementary Data Overall Temporal*

Information, Location & Spatial Information for more information about the specific resources used to geolocate each individual study location.”

Whilst location and time-specific data was available for the CHIRPS rainfall data, the Reviewer is completely correct that for many of the other environmental variables used, the results for each study/location were averaged over time and space (to varying degrees). Per the Reviewer’s suggestion, we have therefore added the following text to the Discussion highlighting this averaging of the covariates used in the multinomial logistic regression:

Discussion (Lines 393-402): *There are a number of limitations to the work presented here – firstly, whilst location and time-period specific data were available for the collated rainfall, varying (often limited) degrees of geospatial information were present in each included study. The environmental covariates used in the multinomial-logistic regression were therefore spatially averaged over reported study area, and additionally often across multiple years due to the absence of time-period specific data. This spatio-temporal averaging may obscure relevant inter-annual variation in factors (e.g. rainfall) that affect population dynamics²³, and may contribute to some of the more limited seasonality (e.g. Clusters 2 and 4) and timing of seasonal peaks (in e.g. Clusters 1 and 3) observed. We mitigate this somewhat by extracting time-period specific rainfall data for each study but cannot preclude some role of spatio-temporal averaging in the results presented here.”*

With regards to their second point, we note that Supplementary Table 3 (previously Supplementary Table 2 in the original manuscript) contains a full list of the covariates considered in the study, and the time-period over which they were averaged (i.e. the temporal resolution).

b) Catch methods: I would state in the methods section the number of different trapping methods used – it need just be a sentence to summarise and then this also needs to be acknowledged in a limitations section in the Discussion. While you normalise the data so that comparisons are relative, this may not account for all biases in differences in trap catches resulting from their different attractiveness for females at different reproductive stages and behaviour, including dispersal, at a given time & influenced by local climate.

We completely agree with the Reviewer’s comment - our work synthesises catches carried out in a diversity of different locations, using a variety of methodologies, timings and (where relevant) a number of different types of bait. We have addressed this comment by adding a sentence to the Methods section (Line 97) as suggested, and then the following text to the Discussion sections of the manuscript’s main text:

Discussion (Lines 402-409): *“Another limitation is the heterogeneity in mosquito sampling methods across the studies. Studies varied in the catch-method used (landing catch, resting collections, pit collections, light traps and spray catches), as well as timing (dawn, dusk, night-time etc) and location (typically either human-dwellings or cattlesheds) of collections. This heterogeneity may interact with mosquito traits (such as timing⁶⁷ or degree of indoor/outdoor biting⁶⁸ and host preferences⁶⁹) that vary between species, and have implications for which species are sampled, and their comparative abundance⁷⁰. We partially mitigate this heterogeneity by normalising the catch data, but this incomplete accounting for differences in catch methodological characteristics might lead to biases in the presented inferences presented.”*

We have also added the following Table and text to the SI:

Supplementary Information (Lines 84-95): *“The studies analysed here employed a wide array of different sampling methodologies including Indoor and Outdoor Resting Collections, Human Landing Catches, Spray Catches and Trap Catches amongst others.*

The majority of studies carried out were resting collections – within each of the different catch methods however, there was further variation according to the location the catch was carried out in (typically human dwellings or cattlesheds), the timing (daytime, night-time or overnight) and (where relevant) the bait used (typically either cattle or humans)

Supplementary Table 2: Summary of the Number of Time Series Collected Using Different Catch Methods. *Note that when summed (260), these values do not correspond to the number of time-series used (272) as in a small number of cases, multiple sampling methods were used, and the results not*

disaggregated (and have therefore not been counted for the purposes of the table above).”

	Landing Catch	Resting Collections	Pit Collections	Light Traps	Spray Catches
# Time Series	41	194	5	15	5

Could you compare the time-series that cluster in either the dry or monsoon peak with those that are perennial and see if there is a correlation with raw catch size? I worry that the lack of seasonality may be due to low catches year-round which could be due to catch method and hence introduce a subtle bias into your analyses and influence the reliability of the conclusions you make from this. Happy to be persuaded otherwise.

We thank the Reviewer for this incredibly insightful comment – we have undertaken a set of analyses exploring total catch-size (for each study) and how the distribution of catch-sizes varies across the 4 archetypes/clusters identified in this work. The results of these analyses are presented in the newly added Supplementary Figure 6. Our results highlight significant variation in total catch sizes between studies, with the extent of this inter-study variation larger than the variation between clusters. Whilst we find some limited differences in catch sizes between Clusters, with Cluster 4 (perennial-like dynamics) having the lowest average (either median or mean) total-catch size, the majority of statistical tests we carried out (t-test for difference in means and Mood’s test for difference in medians) did not support a statistically significant difference in the average total catch size for each cluster.

These findings in-part support our results that the variable degrees of seasonality observed across e.g. Cluster 1 (monsoon-peaking dynamics) and Cluster 2 (bimodal dynamics which are less seasonal) are legitimate and not an artefact arising from disparities in sample size between clusters. However, it is not possible for us to preclude the lack of seasonality in Cluster 4 arising artefactually from the lower catch sizes associated with that Cluster. To that end, we have added the following text describing and highlighting the variation and overdispersion in mosquitoes caught for each study, and how this might impact the results (in particular with respect to the validity and legitimacy of the 4th, “perennial” archetype):

Results (Lines 182-183): *“Average catch size varied between Clusters, ranging from a median catch size of 356 for Cluster 2 to 42 for Cluster 4 (see Supp Fig.4).”*

Discussion (Lines 409-415): *“There were also significant differences in the average number of mosquitoes caught between clusters, with Cluster 4 (perennial dynamics) having the lowest average catch size. Whilst differences in catch sizes between clusters were smaller than within cluster variation (where individual study counts ranged over several orders of magnitude and were highly overdispersed), it is possible that the lack of observed seasonality for Cluster 4 time-series might be an artefact of limited sampling effort and mosquitoes caught.”*

3. Sample size for each species and results

In fourth paragraph of results and with reference to Fig. 3A: *‘in contrast to An. culicifacies s.l. and An subpictus (which clustered together and showed a strong positive assoc. with C1 and -ve with C3), binary indicator for An. dirus displayed weak associations with all clusters, including C1.’*

Looking at Fig. 3A – Fluvialtilis appears to show the strongest correlations (positive for C3 and negative for C1), followed by culicifacies and subpictus, then annularis, stephensi, dirus then minimus.

If we then also look at time-series sample size, fluvialtilis – 60, culicifacies – 85, subpictus – 38, annularis – 39, stephensi – 27, dirus – 11, minimus – 12. Is it possible that the lack of association to a specific cluster is an artefact of small sample size?

In the least this needs to be acknowledged and addressed in the Discussion. Could you sub-sample the larger datasets randomly to simulate what would happen if you only had say 10 time-series for fluvialtilis for example?

We entirely agree with the Reviewer’s comment – utilisation of an L2 penalty within our

multinomial regression-based framework involves placing Bayesian priors centred around 0 on our coefficient estimates, a practice which aids in situations where there is multicollinearity and which stabilises coefficient estimates – it is a commonly used, and widely accepted statistical practice that typically improves model predictive performance. A disadvantage of this approach is that in situations where 1) data is sparse and/or 2) the strength of the association is comparatively weak, there is a more limited shift away from a null hypothesis of no association (i.e. a coefficient value of 0).

In response to the Reviewer's comment, and in consideration of this, we have conducted a series of further analyses involving subsampling of the full dataset. The results of these analyses are presented in Supplementary Figure 7, where we have explored how subsampling the larger datasets (to either 25 or 11 samples per species, with the latter selected to match the lowest sample size for a species i.e. *dirus*) affects our inferences. Specifically, we have:

- Explored how the inferred species-specific coefficients change as a result of subsampling (Supplementary Figure 7 A and B).
- Explore how the results of the hierarchical clustering change in the most extreme subsampling case. (Supplementary Figure 7 C and D)

In Supplementary Figure 7A, we see that for *fluviatilis*, the coefficient values shrink towards 0 as we progressively subsample the data (expected within our framework, where less data means a weaker signal to shift the inference away from a null hypothesis of no association), but that the strongest associations remain significantly larger than those inferred for *dirus*. This shows that inferred associations between *fluviatilis* and cluster membership are not simply a function of more data being available. By contrast however, we see that the inferred coefficient values for *subpictus*, when subsampled to only 11 datapoints, reach similar values to those observed for *dirus*.

These results suggest that the inferences made regarding existence of an association are likely bona-fide (as for *fluviatilis*), but that the apparent absence of associations (e.g. for *dirus*, as assessed by magnitude of the coefficient value) might well be an artefact of low sample sizes.

However, our results in Supplementary Figure 7C and 7D show that the result of hierarchically clustering the coefficient values to identify groups of species with similar patterns of association is largely robust to subsampling, with the exception of *Anopheles dirus*. Based on these results, we have adapted the main text and integrated these findings by reframing the results to shift focus from the actual magnitude of each coefficient value (as previously) to instead focus on the groupings of species with similar patterns of association with different temporal dynamics, which the results of Supplementary Figure 7C & 7D show are (except in the case of *Anopheles dirus*) largely insensitive to the degree of sub-sampling carried out.

Results (Lines 203-218): “Across the species complex regression coefficients, *Anopheles culicifacies* s.l. and *Anopheles subpictus* s.l. demonstrated positive associations with Cluster 1 (monsoon peaking dynamics), whereas for *Anopheles fluviatilis* s.l., this relationship was negative (the species-complex associated with Cluster 3 instead) and *Anopheles annularis* s.l. was most strongly associated with Cluster 4 (perennial dynamics). To explore this variation more systematically, we employed a hierarchical clustering approach to identify groups of species with similar patterns of association with specific temporal dynamics (Fig.3A). *Anopheles culicifacies* s.l. and *Anopheles subpictus* s.l. clustered together and showed a positive association with Cluster 1 and a negative association with Cluster 3). By contrast, *Anopheles fluviatilis* s.l. clustered on its own, positively associated with Cluster 3 and negatively associated with Cluster 1. There were significant disparities in the number of time-series available for each species (ranging from 85 for *Anopheles culicifacies* s.l. to only 11 for *Anopheles dirus* s.l.) and so we explored how robust the results of this clustering were robust to subsampling the data so that all species had the same number of time-series (as *Anopheles dirus* s.l.). Hierarchical clustering showed that these groupings were robust to subsampling, except in the case of *Anopheles dirus* s.l., which instead clustered with *Anopheles culicifacies* s.l. and *Anopheles subpictus* s.l. (and showed positive associations with Cluster 1 dynamics, and a negative association with Cluster 3 dynamics, Supp Fig.6).”

In the Background, references 22 and 23 are used to argue that ‘...*An. funestus* and *An. annularis* populations frequently lack marked seasonal fluctuations in population abundance (22,23 10,24,25). This brings into question how generalisable relationships between rainfall and mosquito population dynamics are’.

I have only looked at references 22 and 23 with respect to *An. funestus*, but these don't support this statement. There is seasonality in *An. funestus* abundance catches as shown in Fig. 1 of Cohuet *et al* (2004) and this generally appears to follow rainfall trends. While >100 individuals were caught at all times of year, there is still seasonality, and I would hypothesise that the perennial transmission was in part due to a relatively short dry season and plenty of rain at other times of year (compare monthly rainfall in this paper with that of Mendis *et al* 2000). For Mendis *et al.* (2000), as shown in their Fig. 2, relatively few *An. funestus* were caught during the entire study and whether sampling was sufficient to be able to detect seasonality is questionable. Here monthly rainfall was usually <50mm and only >150mm twice in c. 1.5yrs. In the Cohuet study rainfall was frequently >100mm and >150mm 8 times. Hence, here the question is being asked whether we can generalise about relationships between rainfall and mosquito population dynamics but the evidence you provide, at least for *An. funestus*, is supportive of a positive relationship between *An. funestus* catches and rainfall.

I would suggest you either provide references that support this statement or change this statement. It might also help if you can be more quantitative in your statements. For example, here ‘*lacked marked seasonal fluctuations*’ could be changed to consider relative differences between species from one or two studies?

We apologise for this – there appears to have been a formatting error with the references (as evidenced by the weird reference order “22,23 10,24,25”) and sentences in the draft intended to be submitted have been lost. We have added these back in, which notes that an array of different dynamics have been observed for *Anopheles funestus* in relation to rainfall and additionally includes a recent reference not available at time of submission showing the population peaking in the dry season in proximity to Lake Victoria:

Matowo, Nancy S., et al. "An increasing role of pyrethroid-resistant Anopheles funestus in malaria transmission in the Lake Zone, Tanzania." Scientific reports 11.1 (2021): 1-13.

We have also adapted the text in response to the Reviewer's suggestions to be more specific about the nature of the dynamics that *Anopheles annularis* displays. Specifically, to note in the cited references that some seasonal fluctuations do occur, but that in contrast to other species, in these references *Anopheles annularis* was caught in significant numbers across the course of the year, including in the dry season when other dominant malaria vectors (such as *Anopheles culicifacies*) are very rarely found.

Minor Comments

Abstract

- ‘Diverse dynamics can be clustered into ‘dynamical archetypes’, each characterised by distinct temporal properties and driven largely by a unique set of environmental factors’

Would suggest remove ‘driven’ and change to stating ‘correlate with’ as you don't show mechanistic relationships between the seasonal dynamics and potential explanatory variables. Would also state that they were clustered into ‘four ‘dynamical archetypes’’. I think a nice finding of the study were that there were just these four clear groups/ patterns.

We have made both sets of changes as suggested by the Reviewer.

- ‘These results highlight that a complex interplay of factors, rather than rainfall alone shape the timing and extent of mosquito population seasonality’

Could you alter slightly to end abstract on more interesting note?! From my perspective, this conclusion is a bit obvious and not terribly exciting. Either potentially something about the

approach and application to other taxa (if it's novel) or the importance of considering 'archetypes' – what would that benefit disease ecology research and vector control?

We have adapted the Abstract and changed the end per the Reviewer's suggestion.

Background

- Although mentioned in the Discussion, I would also briefly summarise here the different larval habitats associated with the different Anophelines if you can afford the word count.

In the interest of brevity, we have not added any further text into the Introduction about larval habitats.

- Following from comment 1 on text which provides motivation for the study, you could consider stating that while many entomological studies have been carried out, they are rarely brought together to be able to make generalisations and to compare and contrast among vector species/ contexts – this was the primary motivation for my interest in the study.

Text to this effect has been added into the Abstract and Introduction per the Reviewer's suggestion.

Results

- Can you be more quantitative throughout the text? E.g. 1st paragraph a) '**substantial variation** in temporal dynamics was observed between different species complexes with **many** of the collated time-series lacking the close, positive correlation with rainfall typically assumed for mosquito populations'; b) 'despite highly seasonal patterns of rainfall, **a number of** time-series belonging to *An. annularis* s.l. demonstrated perennial patterns of abundance'; c) **A number of environmental covariates** also demonstrated cluster-specific associations...'
- You could delete the sentence a) in the first paragraph which would deal with Comment 1 and make this section more succinct. You then follow on to describe this statement in better detail anyway. Also suggest deleting sentence c) as you can go straight into saying this explicitly.

In response to the Reviewer's comments, we have shortened sentence a) significantly and removed sentence c) from the main text, and per their comment immediately below, rephrased b) to be more precise.

- For b) in 10 how many 2? 10? 50? This detail is important – for these studies when were they carried out and have you checked if in the manuscript or elsewhere there are local rainfall data available which may explain the perennial pattern? E.g., back to the averaging question in major comments – it may be that your averaged rainfall is not of sufficient resolution – perhaps these were particularly dry periods.

Per the Reviewer response in the Major Comments above, we note that the rainfall data used for calculating cross-correlations with rainfall is indeed matched to both location and time. However, we agree it would be useful to be more specific and precise about the numbers of time-series being referenced, and what exactly is being referenced. We have therefore added a new sub-figure to Figure 1 (new Figure 1C and 1D) that defines an explicit measure of seasonality (in-keeping with existing definitions of seasonality used in the public-health literature e.g. <https://www.nature.com/articles/ncomms1879>), exploring how concentrated the total annual catch is, specifically what proportion of the total annual catch is concentrated into X months. This provides a more objective measure of difference between the species, which we now describe in the Results section.

In third paragraph '*...indicating different sibling species within a complex display distinct temporal dynamics or that mosquito populations belonging to the species complex are able*

to adopt a diverse array of temporal dynamics depending on the particular ecological setting suggest move to discussion and also change word 'adopt' as make it sound intentional rather than an emergent phenomenon. Similarly, *'This suggests that the ecological features of locations of An. dirus had been sampled in rather than intrinsic to species-complex itself were primary driver of observed dynamics'* move to Discussion.

Addressed per the Reviewer's suggestion.

Figures

Fig 1 – are time series over many different years? While this can be found in the supplementary material and may be mentioned in the methods, would be good to state the years over which data were collected in the legend.

Addressed per the Reviewer's suggestion and text added to Figure 1.

Fig 2 – y axis units different from fig 1, assume fig 1 need to multiply by 100.

Addressed per the Reviewer's suggestion and axes of Figure 1 changed.

Fig 3 – What is CHIRPS?

Per the revisions in response to the major comments above, we have added clarifying information throughout the paper about what exact CHIRPS is.

Discussion

- Third paragraph: *'...our results also support a significant role for the environment in shaping mosquito population dynamics...'*

Be more specific or delete as of course the environment is important in shaping mosquito population dynamics.

Completely agreed – we have deleted this sentence.

- Third paragraph: *'Many of the populations studied here lacked the frequently assumed positive relationship with rainfall and instead displayed patterns of abundance that were only weakly or even negatively correlated. Rainfall is frequently considered a key driver of mosquito population dynamics...'*

'Many' – be quantitative here – at least more specific about which groups? Unless I've misunderstood your analyses don't account for temporal delays in the effect of rainfall? –

e.g. a peak in mosquito abundance often likely doesn't coincide well with the peak in rainfall due to dynamics. I think this should be acknowledged here.

We completely agree with the Reviewer's assessment of this section – we have removed this section from the manuscript.

Reviewer 2

In this paper, the authors take a remarkable number of mosquito population time series data sets from across the Indian sub-continent, and characterize them into clusters of similar dynamics, with respect to seasonal climate. This is a remarkable set of data, and I really enjoyed reading the paper, the temporal dynamics and clustering approaches are well described, and much of the code accessible.

We thank the Reviewer for their useful comments, which have materially improved the manuscript! Below are details about the specific changes we have made in response to their comments (in addition to the other changes requested by the other Reviewer).

My first suggestion is that the title reflect the contents slightly better, and include the geographic focus – perhaps add ‘across the Indian subcontinent’ or similar. This both gives a picture of the sheer breadth of area covered in the paper, AND, distinguishes this from other more regionally specific malaria papers (which the authors allude to). I think this will draw attention to the assemblage of species nicely, and allows for researchers from other regions to appreciate the difference.

We apologise for the confusion here – the systematic review was carried out including only studies conducted in the country of India – we have removed all references to the Indian subcontinent to clarify this. We respectfully disagree with the suggestion to change the title – part of the study is aimed at characterising the patterns of mosquito population dynamics across India, but a significant novelty of this work is presentation of the methodology for clustering time-series into groups with similar temporal dynamics – this methodology is broadly applicable beyond the work presented here, both with respect to geography and the mosquito species considered, and so we would suggest retaining the title as it is currently.

My second suggestion is that since one of the major conclusions is that the population dynamics are far less governed by rainfall than expected, I would draw attention to the papers examining temperature (thermal boundaries and optima) as a driver of mosquito-borne disease transmission – in particular, (Miazgowicz et al. 2020) address this in *An. stephensi*, one of the more urban mosquitoes in the Asian study region. There are several preceding papers, examining malaria and malaria vectors, both describing the thermal curves, and mapping the results (Johnson, Lafferty, et al. 2015; Johnson, Ben-Horin, et al. 2015; Ryan et al. 2015; Mordecai et al. 2013), and a preprint of work in review (Villena et al. 2020). The authors touch on the issue that rainfall can be complicated to use as a driver, given how rainfall translates into breeding habitat, and that is another point that would be great to emphasize further, and perhaps offer some pointers to researchers in other regions (e.g. the Americas) who are approaching these questions; given the number of different habitats within the Indian subcontinent, this would be very helpful. I can see that this spatio-temporal population clustering approach would apply to many regions and multiple VBD systems.

We thank the Reviewer for this thoughtful comment and the comprehensive set of suggested references. We agree with their assessment about the importance of the role of temperature in driving these dynamics and have added both further text and the suggested references into the revised manuscript:

Discussion (Lines 352-361): *“The role of temperature in shaping mosquito population dynamics is increasingly being recognised^{29,51}, due in part to the significant influence it has on many individual mosquito life-history traits^{27,52,53}, including biting rate, fecundity and mortality (amongst others); with its influence on these factors typically non-linear and unimodal with clear optima⁵⁴ and subject to interactions with other factors such as the demographic structure of the mosquito population⁵⁵. Together, this has significant consequences for mosquito population dynamics and, in turn, the range and dynamics of vector-borne diseases (such as malaria) they underpin⁵⁶⁻⁵⁸. Our results therefore suggest a role for both rainfall and temperature in shaping annual patterns of mosquito abundance and underscores the importance of considering seasonal fluctuations in a range of environmental variables when trying to understand seasonality in mosquito population dynamics.”*

In Figure 1, would it be feasible to add a bit more geographic information? The first panel

could benefit from perhaps some context (label India, perhaps point out the administrative level being depicted by the lines, add country labels to the north, and maybe even a simple relief map as the backdrop, rather than yellow; give a scale bar). One can leverage a map to convey a LOT to readers, and given the limit on figures in articles, this is a good opportunity to do that. I would also make the internal black lines grey, and check the dot colors for colorblind contrast.

We apologise for the confusion – the map in Figure 1 is just of India (not other countries). We have added text into the Figure legend to address this and provide further information to the readers – we have also checked the dot colors for colorblind contrast.

Figure 2 – if the panel layout is as in the review copy, I would put the Cluster descriptors (Monsoon, Bimodal, etc) as call out labels on the ellipses, OR, label the ellipses as Clusters 1-4, and place the labels in B as well. I really like Figure 2C, as a means to see the species communities within the clusters.

We completely agree with the Reviewer's assessment here and have incorporated all of their suggestions into Figure 2, which has also been significantly altered in response to the comments of Reviewer 1.

Figure 4 and the use of the predictive maps – I felt like this piece of the paper rather didn't entirely hang with the rest of it; the predictive mapping is part of another project, and the website they are on is less accessible than the data and code within this paper. The probability maps are generated using a different type of modeling, and the reliability of the output not fully described up front. I think either I would have liked to see multiple niche-generating algorithm output maps, to show the range of importance and geographic spread, or to have this piece better integrated into the main body of the paper, rather than referring the reader to a supplement elsewhere for the methods. While this is not a dealbreaker, it just sits oddly for me. In the figure, the predictive map is just presented as a given probability gradient – these are notoriously hard to interpret, and thus I would maybe use a probability cutoff method instead (e.g. >0.5), to better represent where the reader should visually cue in on the fit.

We thank the Reviewer for their thoughtful comments on this section of the manuscript. We feel that the inclusion of the predictive mapping is useful as an illustrative example of how this new analytical framework (which generates predictions of the temporal dynamics associated with specific mosquito populations) might be integrated into existing spatial analyses (of mosquito presence/absence, which have dominated the entomological and ecological literature to date). This is in agreement with Reviewer 1 who noted this as an area of particular interest: *"The study is novel, and I think the approach, which combines both temporal and geospatial analyses, will be of interest to entomologists and more generally ecologists."* To that end, we have elected to retain the predictive mapping results, but completely agree with the Reviewer's assessment that the figure is difficult to interpret and so have implemented the idea of thresholding they have suggested (at $p>0.67$) to aid interpretability and highlight the points of overlap/difference in the geographical extent and range of the different temporal dynamics.

I have no minor edits to list out here, but would happily see a revised version addressing the rearrangement of text in response to my suggestions, and updated figures.

Data and Code Access: On the GitHub repository, the labelling suggests this was a PNAS submission, and the link to 'data' in the instructions to download, does not work. This means the data are actually NOT accessible beyond the copy appended with the reviewer proof. I would recommend the authors correct this, and perhaps relabel/rename the repository (or check that the right one is linked in the paper).

We apologise for this oversight – we, of course, intend for the data to be freely available, and every single analysis carried out and displayed in this study to be absolutely reproducible. We have updated the GitHub repository (<https://github.com/cwhittaker1000/anopheleseasonality>) accordingly so that it now contains

all the data required to reproduce the analyses presented. We have also organised a GitHub “release” of the repository, integrated with the archival platform Zenodo to produce a persistent, archived version of the release with a Digital Object Identifier (DOI) – links to all of these are available via the GitHub repository.

Overall, I think this paper presents a useful set of methods applied to a remarkable set of data, and it will be a nice contribution to the literature, and to the toolkit for vector borne disease analyses. For this latter reason, I would really like to see a clean repository and working data access.

- Johnson, Leah R., Tal Ben-Horin, Kevin D. Lafferty, Amy McNally, Erin Mordecai, Krijn P. Paaijmans, Samraat Pawar, and Sadie J. Ryan. 2015. “Understanding Uncertainty in Temperature Effects on Vector-Borne Disease: A Bayesian Approach.” *Ecology* 96 (1): 203–13.
- Johnson, Leah R., Kevin D. Lafferty, Amy McNally, Erin Mordecai, Krijn P. Paaijmans, Samraat Pawar, and Sadie J. Ryan. 2015. “Mapping the Distribution of Malaria: Current Approaches and Future Directions.” In *Wiley Series in Probability and Statistics*, 189–209. Hoboken, NJ, USA: John Wiley & Sons, Inc.
- Miazgowicz, K. L., M. S. Shocket, S. J. Ryan, O. C. Villena, R. J. Hall, J. Owen, T. Adanlawo, et al. 2020. “Age Influences the Thermal Suitability of Plasmodium Falciparum Transmission in the Asian Malaria Vector Anopheles Stephensi.” *Proceedings. Biological Sciences / The Royal Society* 287 (1931): 20201093.
- Mordecai, Erin A., Krijn P. Paaijmans, Leah R. Johnson, Christian Balzer, Tal Ben-Horin, Emily de Moor, Amy McNally, et al. 2013. “Optimal Temperature for Malaria Transmission Is Dramatically Lower than Previously Predicted.” *Ecology Letters* 16 (1): 22–30.
- Ryan, Sadie J., Amy McNally, Leah R. Johnson, Erin A. Mordecai, Tal Ben-Horin, Krijn Paaijmans, and Kevin D. Lafferty. 2015. “Mapping Physiological Suitability Limits for Malaria in Africa Under Climate Change.” *Vector Borne and Zoonotic Diseases* 15 (12): 718–25.
- Villena, Oswaldo C., Sadie J. Ryan, Courtney C. Murdock, and Leah R. Johnson. 2020. “Temperature Impacts the Transmission of Malaria Parasites by Anopheles Gambiae and Anopheles Stephensi Mosquitoes.” *Cold Spring Harbor Laboratory*. <https://doi.org/10.1101/2020.07.08.194472>.

Appendix D

The authors have more than addressed my comments from the first round of review. This is a great paper and would be good to see it published in Proc B. Please just find some very minor suggestions below, mainly on wording.

37: Would delete 'a feature that results in' and replace with 'with'.

42: Don't agree – can you say beyond a focus on dynamics in single locations?

45-47: Would suggest get rid of parentheses - Understanding the determinants of these dynamics is important given that the efficacy of interventions including malaria chemoprevention^{11,12} and indoor-residual spraying^{13,14} depends...

55, 56: Here and elsewhere they are not breeding sites but larval habitats.

56: Rather than 'studies' show, would prefer to state authors e.g. X and Y et al. show...

58-61: Possibly shorten to: Relatedly, a number of studies have demonstrated *Anopheles annularis* *s.l.* is detected in significant numbers over the course of the entire year despite highly seasonal rainfall.^{25,26}.

68-72: Again, simplify sentence here by just stating what is in the parentheses – there is quite a lot of parentheses throughout the text which can interrupt flow – good if you can get rid of at least some.

73-74: Would just delete first sentence.

213-214: 'robust' is used twice remove the latter

312: thresholder? Typo?

328: two full stops

327: again could remove parentheses here

340-346: long sentence try and split

353-356: another long sentence

367&374: more 'larval' habitats.

Appendix E

Below is a point-by-point list of how we've addressed these new of comments from the Reviewers, but I also just wanted to say a brief but huge thank you to the Reviewers for all their suggestions and comments on the manuscript to date. As an early career researcher with limited experience in navigating the submission and peer-review process, it genuinely is so lovely to get helpful and insightful comments that materially improve the submitted work (and they unequivocally have) – so thank you much again for taking the time to go through and review the work in so much detail! Thank you – it's hugely appreciated!

Reviewer 1

The authors have more than addressed my comments from the first round of review. This is a great paper and would be good to see it published in Proc B. Please just find some very minor suggestions below, mainly on wording.

37: Would delete 'a feature that results in' and replace with 'with'.

Addressed and deleted per the Reviewer's suggestion. Section now reads:

“Transmission occurs via mosquito vectors belonging to the Anopheles genus – these vectors are heterogeneously distributed across the globe^{5,6}, with marked differences in the transmission dynamics of malaria across different ecological contexts.”

42: Don't agree – can you say beyond a focus on dynamics in single locations?

Addressed – have added in “beyond a focus on dynamics in single locations” per the Reviewer's suggestion. Section now reads:

“By contrast, beyond a focus on dynamics in single locations, comparatively less attention has been paid to understanding the temporal patterns of vector abundance, and how these dynamics are shaped by the local environment.”

45-47: Would suggest get rid of parentheses - *Understanding the determinants of these dynamics is important given that the efficacy of interventions including malaria chemoprevention^{11,12} and indoor-residual spraying^{13,14} depends...*

Addressed – have removed the parentheses per the Reviewer's suggestion.

“Understanding the determinants of these dynamics is important given that the efficacy of many interventions including seasonal malaria chemoprevention^{11,12} and indoor-residual spraying^{13,14} depends on the timing of their delivery in relation to seasonal peaks in risk.”

55, 56: Here and elsewhere they are not breeding sites but larval habitats.

We have addressed all the instances in the text where “breeding sites” were referred to, and have replaced them with references to the habitat.

56: Rather than 'studies' show, would prefer to state authors e.g. X and Y et al. show...

Addressed – have explicitly noted the work by “Cohuet et al and Mendis et al” in the section the Reviewer refers to. Section now reads:

“However, a number of studies including work by Cohuet et al and Mendis et al of Anopheles funestus s.l. populations have identified varying degrees of seasonality^{22,23} including population abundance peaking in the dry season²⁴.”

58-61: Possibly shorten to: *Relatedly, a number of studies have demonstrated Anopheles annularis s.l. is detected in significant numbers over the course of the entire year despite highly seasonal rainfall,^{25,26}*

Addressed – have shortened per the Reviewer's suggestion. Section now reads:

“Relatedly, a number of studies have shown Anopheles annularis s.l. present in significant numbers over the course of the entire year, despite highly seasonal rainfall^{10,25,26}.”

68-72: Again, simplify sentence here by just stating what is in the parentheses – there is quite a lot of parentheses throughout the text which can interrupt flow – good if you can get rid of at least some.

Addressed – have removed the parentheses and also broken up the longer sentence into several shorter ones.

73-74: Would just delete first sentence.

Addressed, have deleted the first sentence here.

213-214: 'robust' is used twice remove the latter

Addressed, have removed the second usage of robust in that sentence.

312: thresholder? Typo?

Addressed – this should be “*thresholded*”, an error which we have now rectified.

328: two full stops

Addressed, thanks to the Reviewer for spotting this!

327: again could remove parentheses here

Addressed, parentheses removed and sentence simplified. Section now reads:

“Analysis of this variation has revealed a complex interplay between species complex-specific drivers and the broader ecological structure of the environment in shaping these dynamics.”

340-346: long sentence try and split

Addressed, have split the long sentence in two. Section now reads:

*“An important limitation to note however is our inability to disaggregate time-series according to sibling species, which frequently show differences in preferred types of aquatic larval habitat⁵⁰. It therefore remains unclear whether the variation in temporal dynamics for *Anopheles culicifacies* s.l. is due to sibling species displaying distinct temporal dynamics or because *Anopheles culicifacies* s.l. temporal dynamics are generally more plastic than *Anopheles fluviatilis* s.l. (where the same dynamics were typically observed irrespective of the broader ecological structure).”*

353-356: another long sentence

Addressed, have split into separate sentences. Section now reads:

“The role of temperature in shaping mosquito population dynamics is increasingly being recognised^{9,51}, due in part to the significant influence it has on many individual mosquito life-history traits^{27,52,53}, including biting rate, fecundity and mortality (amongst others). The influence of temperature on these traits is typically non-linear and unimodal with clear optima⁵⁴ and subject to interactions with other factors such as the demographic structure of the mosquito population⁵⁵.”

367&374: more 'larval' habitats.

Addressed, all instances in the text of “breeding sites” have been changed to refer to larval habitats, per the Reviewer’s suggestion.

Reviewer 2

I appreciate the authors' attention to the reviewer comments, and the responses are largely adequate. I really appreciate the clean GitHub repo and attention to detail - it might be useful to connect to a service such as Dryad for storage of larger files, so that the project doesn't deprecate and require future researchers to contact the authors about shapefiles, e.g. This is just a suggestion for making the project a useful package, rather than a necessary fix at this stage. If the intent is for future users to use the tools and be able to see the full implementation, being able to locate the data easily is part of that practice, but I also appreciate it's beyond the scope of journal obligation.

We appreciate the Reviewer's kind words and appreciate them flagging this issue in their original set of comments, which has now ensured everything is fully accessible! We agree entirely about making sure the larger files (and the contents of the repository more generally) are accessible in perpetuity. The integration with Zenodo means that the current version of the repository (i.e. all analysis scripts, datasets etc) all exist in an openly accessible repository via this link: <https://zenodo.org/record/5862952/>. Importantly, this link is static (i.e. represents a snapshot of the repository and its current contents) and has an associated DOI (DOI: [10.5281/zenodo.5862952](https://doi.org/10.5281/zenodo.5862952)) that will remain active and accessible irrespective of any future modifications to the repository – together with the link to the version of the Github release (<https://github.com/cwhittaker1000/anopheleseasonality/releases/tag/v1.0.3>) available on the repo (which contains static versions of the larger files), this means the exact contents of the repo as it currently stands will therefore remain accessible to the research community in perpetuity.

Additionally, we also agree entirely with the Reviewer that it's important not only for these links to exist, but also that part of truly making things open involves making them easily findable and accessible. To that end, we have added both the Zenodo and Github links mentioned above to the manuscript signposting researchers to these static versions, and also new text to the Github README, showing researchers where **exactly** within the Datasets subdirectory the collated entomological data resides. Together, we hope this will facilitate increased ease of access to this work and the data underlying it.

I now really like Figure 2, and am glad that my minor suggestions give it a bit more cohesion, and I think it really sings.

Thank you for the kind words! We also feel like Figure 2 is much improved compared to the original, so thank you again for your suggestions on it!!

I would like to see a justification for the probability cutoff for the spatial mapping component - the word thresholded is misspelled in the figure legend (minor correction). I wasn't sure where the choice of 0.67 comes from, and I would like to know that 2/3 majority has meaning in the context of predicting presence. Perhaps the authors could provide evidence from an external validation exercise, or similar.

We thank the Reviewer for their comment – we chose 0.67 as an arbitrary cut-off and agree, given this, there should be additional text describing this (and results contextualising this), which we have added to the main text. Whilst we feel that an external validation exercise is beyond the scope of the work presented here, we agree the arbitrary nature of the current threshold is not currently clear and should be highlighted. To that end we have generated versions of the maps displaying the actual probabilities for each temporal archetype and pixel. We enclose these figures in the Supplementary material (new Supplementary Figure 9), and reference them with new text in the Results section and Figure 4 legend; and also add additional text highlighting the arbitrary nature of the threshold used for the figure (Figure 4) presented in the main text.

Lastly, I still think the title should be clear that this is India, or, if the intent is, as the authors response suggests, to introduce a novel set of tools, make the title more indicative of that aspect instead. I want to emphasize that it is very useful to signpost that these are novel methods and approaches, so that future researchers will apply them and be able to challenge them with mosquito (and other species') population trends from other locations.

We understand the point the Reviewer raises – and agree with them that it would be useful to adapt the title to highlight the novelty of the developed tools and their applicability to data

from other mosquitoes/species more generally (as an aside, this general point which the Reviewer raised previously about emphasising the novelty of the framework more generally was incredibly helpful and we feel has broadened the scope of the paper and who it is of interest to, so thank you again for those prior suggestions)! We have therefore amended the title, which is included in the version of the manuscript associated with this resubmission. The proposed new title is "*A Novel Statistical Framework for Exploring the Population Dynamics and Seasonality of Mosquito Populations*".

In summary, I am very keen on this paper, it is still a pleasure to read, and I look forward to seeing the final version.

Thank you for the kind words and useful comments!